# Exposure to formaldehyde and asthma outcomes: A systematic review, meta-analysis, and economic assessment

Juleen Lam[1,2], Erica Koustas[3], Patrice Sutton[1], Amy M. Padula[1], Michael D. Cabana[4,5], Hanna Vesterinen[3], Charles Griffiths[6], Mark Dickie[7], Natalyn Daniels[1], Evans Whitaker[5], Tracey J. Woodruff[1]*

1 University of California San Francisco, Program on Reproductive Health and the Environment, San Francisco, California, United States of America, 2 Department of Health Sciences, California State University, East Bay, Hayward, California, United States of America, 3 Scientific Consultant to the University of California, San Francisco, California, United States of America, 4 University of California San Francisco, Philip R. Lee Institute for Health Policy Studies, San Francisco, California, United States of America, 5 University of California San Francisco, Schools of Medicine and Pharmacy, San Francisco, California, United States of America, 6 U.S. Environmental Protection Agency, National Center for Environmental Economics, Washington, DC, United States of America, 7 Department of Economics, University of Central Florida, Orlando, Florida, United States of America

* tracey.woodruff@ucsf.edu

**Data Availability Statement:** All relevant data are within the manuscript and its Supporting information files.

## Abstract

### Background

Every major federal regulation in the United States requires an economic analysis estimating its benefits and costs. Benefit-cost analyses related to regulations on formaldehyde exposure have not included asthma in part due to lack of clarity in the strength of the evidence.

### Objectives

1) To conduct a systematic review of evidence regarding human exposure to formaldehyde and diagnosis, signs, symptoms, exacerbations, or other measures of asthma in humans; and 2) quantify the annual economic benefit for decreases in formaldehyde exposure.

### Methods

We developed and registered a protocol in PROSPERO (Record ID #38766, CRD 42016038766). We conducted a comprehensive search of articles published up to April 1, 2020. We evaluated potential risk of bias for included studies, identified a subset of studies to combine in a meta-analysis, and rated the overall quality and strength of the evidence. We quantified economics benefit to children from a decrease in formaldehyde exposure using assumptions consistent with EPA's proposed formaldehyde rule.

### Results

We screened 4,821 total references and identified 150 human studies that met inclusion criteria; of these, we focused on 90 studies reporting asthma status of all participants with

**Funding:** Funding source: JPB Foundation (Grant #681), NIEHS P01ES022841, USEPA RD 83543301. The funders had no role in study design, data collection and analysis, decision to publish, or preparation of the manuscript.

**Competing interests:** The authors have declared that no competing interests exist.

quantified measures of formaldehyde directly relevant to our study question. Ten studies were combinable in a meta-analysis for childhood asthma diagnosis and five combinable for exacerbation of childhood asthma (wheezing and shortness of breath). Studies had low to probably-low risk of bias across most domains. A 10-µg/m$^3$ increase in formaldehyde exposure was associated with increased childhood asthma diagnosis (OR = 1.20, 95% CI: [1.02, 1.41]). We also found a positive association with exacerbation of childhood asthma (OR = 1.08, 95% CI: [0.92, 1.28]). The overall quality and strength of the evidence was rated as "moderate" quality and "sufficient" for asthma diagnosis and asthma symptom exacerbation in both children and adults. We estimated that EPA's proposed rule on pressed wood products would result in 2,805 fewer asthma cases and total economic benefit of $210 million annually.

## Conclusion

We concluded there was "sufficient evidence of toxicity" for associations between exposure to formaldehyde and asthma diagnosis and asthma symptoms in both children and adults. Our research documented that when exposures are ubiquitous, excluding health outcomes from benefit-cost analysis can underestimate the true benefits to health from environmental regulations.

## Introduction

Formaldehyde exposure is ubiquitous and occurs in homes, communities, and workplaces. Formaldehyde is a high-volume production chemical with numerous industrial and commercial uses as a solution, disinfectant, preservative or to produce industrial resins used to manufacture adhesives and binders in wood, paper, and other products. It is present in many household products, such as foam insulation, cleaning and personal care products, pressed wood products such as particleboard and plywood, and as a result is a common indoor air pollutant found in virtually all homes and buildings [1–9]. Homes are impacted by off-gassing of formaldehyde from new housing materials, with availability and rates of ventilation having minimal impact on exposure levels [10].

In particular, formaldehyde is an environmental justice and affordable housing concern. Lower-income communities are disproportionately at risk of exposure to formaldehyde and resulting health effects from pressed wood products in homes built with less costly building materials. Formaldehyde exposure extends beyond residential homes—for instance, formaldehyde has been measured at levels exceeding exposure limits in childcare settings in California. Workplace exposure to formaldehyde occurs in a wide variety of industries and occupations, such as in the manufacture or production of formaldehyde or formaldehyde-based products or during firefighting, embalming, carpentry, and pathology lab work.

Asthma is a complex disease caused by chronic inflammation of the airways that results in episodic airway hyper responsiveness, excessive mucous secretion, and airway obstruction. Exposure to formaldehyde occurs primarily through inhalation and also as a respiratory contact irritant [11]. The relationship between exposure to formaldehyde and asthma has been actively under evaluation by government agencies for the last few decades [12–14]. A substantial amount of research exploring relationships between formaldehyde exposure and exacerbation of asthma has been conducted, but few systematic reviews (with a pre-established

Timeline of EPA formaldehyde assessment

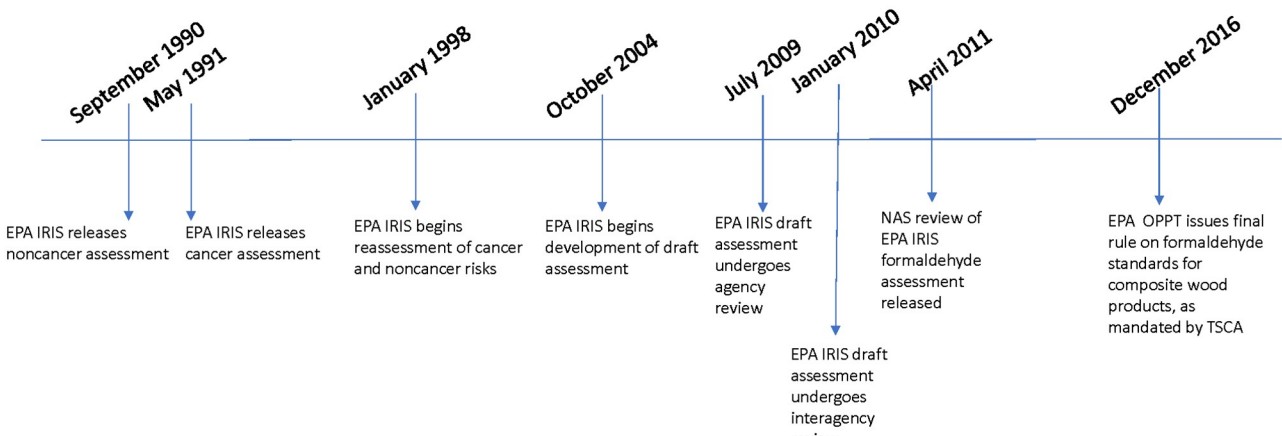

**Fig 1. Timeline of U.S. Environmental Protection Agency (EPA) action on formaldehyde from September 1990-December 2016, highlighting Integration Risk Information System (IRIS) final assessments releases, reassessments, internal and external reviews, and final rules issued.**

protocol, systematic literature search, pre-defined criteria for evaluating studies and categories to assess the strength of evidence) are available providing a comprehensive overview of the evidence.

The U.S. Environmental Protection Agency (EPA) released its review of formaldehyde health risks in its Integrated Risk Information System (IRIS) assessment in 1990, initiated a reassessment in 1998, and released a draft report in 2010, which included a review of the asthma health outcome (Fig 1). A review of the draft assessment by the National Academy of Sciences (NAS) highlighted many methodological limitations of the IRIS process, such as EPA's study selection and evaluation criteria that led to the advancement of one study [15] with potential misclassification of infection-associated wheezing in young children as asthma [14]. EPA's conclusion of a causal relationship between formaldehyde exposure and asthma incidence and subsequent derivation of a candidate Reference Concentration (RfC) was ultimately challenged by the NAS committee [14].

In 2010, Congress required EPA to issue a rule on pressed wood products and emissions of formaldehyde; ultimately EPA issued a final rule on formaldehyde in 2016 (Fig 1). EPA conducted a benefits cost analysis of this rule under an Executive Order that requires every significant regulation in the U.S. be accompanied by an economic analysis of the benefits and costs of implementation. EPA initially included asthma in the benefit-cost analysis for the proposed rule; however, asthma was removed from the analysis after interagency review. In the U.S., asthma affects approximately 23 million people, including 6 million children [16], impacting approximately 8% of both children and adults [17]. The omission of asthma from the benefit-cost analysis could significantly underestimate the true value of regulating formaldehyde in pressed wood products.

To assess the evidence of formaldehyde's contribution to asthma outcomes, we conducted a systematic review of human studies to answer the question of whether exposure to formaldehyde is associated with diagnosis, signs, symptoms, exacerbation, or other measures of asthma in humans. We used results from the quantitative evaluation of the evidence to estimate the

benefits of the reduction in asthma cases implied by the proposed EPA rule on pressed wood products.

## Methods

We applied the Navigation Guide systematic review methodology, a systematic and transparent method for synthesizing the available scientific evidence designed specifically for environmental exposures [18,19]. The method is based on Cochrane and GRADE methods [20,21] and includes the same elements (protocol development, risk of bias evaluation, evidence evaluation, etc.). However, one main difference is that this method accounts for differences in evidence and decision context inherent to environmental health assessments, i.e., the reliance on human observational studies in the absence of randomized controlled trials (RCTs), and the fact that population exposure to exogenous chemicals precedes evidence of their safety.

### Protocol

We developed a protocol prior to initiating the review and registered it in PROSPERO in May 2016 (http://www.crd.york.ac.uk/PROSPERO/; Record ID #38766, CRD 42016038766).

### Study question

Our systematic review objective was to answer the question: "Is exposure to formaldehyde associated with the diagnosis, signs, symptoms, exacerbation, or other measures of asthma in humans?"

The "Participants", "Exposure," "Comparator" and "Outcomes" (PECO) statement is briefly outlined below, with additional specifics available in the protocol.

**Participants.** Humans.

**Exposure.** Any indoor or outdoor sources of airborne inhalation exposure to formaldehyde, including but not limited to occupational, outdoor ambient, indoor household settings, and/or exposure to household products that occurred prior or concurrent to health outcome.

**Comparator.** Humans exposed to lower levels of formaldehyde than the more highly exposed humans.

**Outcomes.** Any of the following asthma-related outcomes: diagnosis of asthma, asthma signs or symptoms, asthma exacerbation, or indirect measures of asthma.

### Data sources

We searched the databases PubMed, ISI Web of Science, Biosis Previews, Embase, Google Scholar, and Toxline from the inception of each database up to April 1, 2020 using the search terms in S1–S5 Tables. We did not limit our search by language or initial publication date. We used the Medical Subject Headings (MeSH) database to compile synonyms for formaldehyde and asthma-related outcomes. Our search terms and search strategy were developed by two librarians trained in systematic review methodology (LS, EW). We also supplemented these results by searching toxicological and grey literature databases (S6 and S7 Tables), consulting with subject matter experts, and hand-searching references by reviewing reference lists of included studies and review papers on the topic as well as searching for references that cited included studies ("snowball searching").

### Study selection

We included studies that contained original data from human studies that measured or reported formaldehyde exposure prior to evaluating the health outcome. We screened

references for inclusion using structured forms in DistillerSR (Evidence Partners; available at: http://www.systematic-review.net). Two of four possible reviewers (EK, ND, AP, HV) independently reviewed titles and abstracts of each reference to determine eligibility in a non-random assignment (to ensure that the same two authors did not always screen the same references). In the event that an abstract was missing or there were discrepancies between the two reviewers, the default was to move the reference forward for full text review. Two of the same four reviewers (EK, ND, AP, HV) then independently performed a full-text review to evaluate inclusion criteria of each reference not excluded by title/abstract screening. An additional reviewer (JL) screened five percent of the titles/abstracts and full-texts for quality assurance.

We excluded studies if any one of the following criteria was met: 1) the report did not contain original data; 2) the article did not involve human subjects; 3) there was no report of formaldehyde exposure; 4) there was no report of diagnosis of asthma, asthma signs or symptoms, asthma exacerbation, or indirect measures of asthma (such as daily use of inhaler); or 5) there was no comparator—control group or exposure range comparison (S1 Methods). We translated the title and abstracts of studies using freely available online software (i.e., Google Translate) that were not published in English to evaluate its relevance.

## Data extraction

We extracted data from studies in duplicate in a Health Assessment Workplace Collaborative database (HAWC; available at: https://hawcproject.org/about/). Two of three possible extractors (SE, EM, DB) independently extracted data relating to study characteristics and outcome measures (S2 Methods) from each included article. A third extractor (PH, BV) performed QA/QC on all the studies to resolve any discrepancies between the two independent extractors; subsequently, two authors (JL, EK) reviewed all studies to further ensure the accuracy of extracted data. When information was missing from a published article, we contacted corresponding study authors to request additional information.

## Rate the quality and strength of the evidence

**Statistical analyses.**  Prior to study selection, we developed a list of study characteristics (contained in our protocol: http://www.crd.york.ac.uk/PROSPERO/; Record ID #38766, CRD 42016038766) to identify studies suitable for meta-analysis. After evaluating the characteristics of all the studies, we grouped studies into four study population and health outcome combinations: 1) child asthma diagnosis; 2) child asthma exacerbation and symptoms; 3) adult asthma diagnosis; and 4) adult asthma exacerbation and symptoms.

To differentiate child from adult studies, we initially planned to use the age of 18 years as a cutoff for children, but a number of the studies used a cutoff age of 15 years to distinguish between children and adults. Given that the onset of asthma commonly occurs during preschool years and recent increases in asthma incidence over the past few decades has been observed to increasingly affect children and adolescents aged 1 to 14 years, we decided to use age 15 years as the cutoff to group child vs. adult studies. We did not include studies in the meta-analysis that reported effect estimates with only mixed children and adult populations in the meta-analysis due to concerns that differences in adult-onset versus childhood-onset of asthma would be masked. We also did not consider these data in our overall rating of study quality and strength, but we did include these data in visual scatterplots of data for comparison with child and adult data.

For the adult studies, we considered the body of evidence to include all adult population studies, regardless of whether exposure occurred in the general population or at work, as

biologically, the relationship between exposure and health outcome is independent of where the exposure occurred. We distinguished the adult general population study results from the adult occupational study results on the visual scatterplots for comparison.

For cohorts with multiple publications (for instance, if a cohort was followed over time), we utilized results from the latest time point where our relevant outcome of interest was measured, but also considered information provided collectively across the publications for evaluating study quality. Where available, we used adjusted odds ratios to conduct the meta-analysis but if adjusted results were not reported, we included unadjusted ORs in the analyses. We converted effect estimates to an OR and 95% confidence interval (CI) for the association between asthma per 10-$\mu g/m^3$ unit increase in formaldehyde exposure to standardize across studies, transforming units of exposure when necessary. Where a meta-analysis was not possible, we created visual scatterplots of data across studies reporting on similar outcomes and subpopulations to consider all available data in assessing the evidence. We also applied a mixed models approach for repeated data to evaluate outcomes at various doses, using exchangeable correlation structures for repeated measurements within the same study.

We evaluated statistical heterogeneity across study estimates in the meta-analysis using $I^2$ with $p \leq 0.05$ as our cut off for statistical significance, as previously described. If statistical heterogeneity was present, we used leave-one-out analysis to identify the study or studies contributing, evaluated potential study characteristics (e.g., study location, study population, study design, adjusted confounders, timing of exposure, etc.) to determine if we could explain the source, and incorporated hierarchical cluster structures in the data analysis to statistically account for heterogeneity. We also investigated the relative contribution of each study to the overall meta-analysis association and conducted sensitivity analysis to investigate the impacts of removing highly influential studies from the analysis. Data management was performed with Microsoft Excel. Statistical analyses were performed using STATA 13.1 software (Stata-Corp, 2011). We pooled estimates using inverse variance-weighted models, fixed-effects models and the DerSimonian and Laird random-effects models. We used the *metan*, *metareg*, *metainf*, *metafunnel*, *metabias* and *metatrim* packages in STATA version 13.1.

To investigate the effect of publication bias on our meta-analysis, we created funnel plots and used Egger's test. We also quantitatively evaluated each meta-analysis for the potential effect that a new study might have on changing the interpretation of our overall results. Specifically, the association estimate of a new or unpublished study necessary to alter the results of the meta-analysis was calculated under two scenarios: 1) the 95% confidence interval of the meta-analysis overlapped zero, and 2) the meta-analysis central association estimate was greater than zero (moved to the opposite direction—i.e., such that increases in formaldehyde exposures would be associated with decreases in asthma outcomes). In making this calculation, we assumed that the new hypothetical study would have a standard error equal to the smallest in our group of studies.

**Assessing the risk of bias for each included study.** We evaluated risk of bias separately for each of the four study population/outcome group combinations using The Navigation Guide Risk of Bias Tool, a modified instrument based on the Cochrane Collaboration and Agency for Healthcare Research and Quality (AHRQ) domains, with customized instructions for each domain based on the type of evidence anticipated beforehand (S3 Methods).

We evaluated nine risk of bias domains (Source Population, Blinding, Outcome Assessment, Confounding, Incomplete Outcome, Exposure Assessment, Selective Reporting, Financial Conflict of Interest, and Other). We assigned each domain as "low," "probably low," "probably high," or "high" risk of bias, or "not applicable" (domain not applicable to study) according to specific criteria as described in our risk of bias instruments (S3 Methods). Two of three possible reviewers (SE, EM, RB) independently recorded risk of bias determinations for

each included study. We held an in-person meeting for all review authors (JL, EK, PS, AMP, MDC, HV, ND, EW, TJW) to review risk of bias ratings and rationales for each study, come to consensus to ensure consistency, and record our final rationale. One review author (EK) independently reviewed all final risk of bias ratings for QA/QC.

**Rating the quality of evidence across all included studies.** We separately rated the quality of the overall body of evidence as "high," "moderate," or "low" for each of the four study population/outcome group combinations. We assigned an initial rating of "moderate" quality for each group of human observational studies prior to evaluating the included studies, based on previously described rationale—briefly, observational human studies are recognized as a reliable source of evidence and generally the most appropriate for answering environmental health-related questions. From the initial "moderate" quality rating, we then considered potential adjustments ("downgrades" or "upgrades") to the quality rating based on 8 categories of considerations: risk of bias, indirectness, inconsistency, imprecision, potential for publication bias, large magnitude of effect, dose response, and whether residual confounding would minimize the overall effect estimate; the specific factors and criteria considered are outlined in S4 Methods. Possible ratings were 0 (no change from initial quality rating), -1 (1 level downgrade) or– 2 (2 level downgrade), +1 (1 level upgrade) or +2 (2 level upgrade). Review authors independently evaluated the quality of the evidence and then we compared ratings as a group and recorded the consensus and rationale for each decision.

**Rating the strength of the evidence across all included studies.** We assigned an overall strength of evidence rating separately for the four study population/outcome group combinations based on four considerations: (1) Quality of body of evidence (i.e., the rating from the previous step); (2) Direction of effect; (3) Confidence in effect (likelihood that a new study would change our conclusion); and (4) Other compelling attributes of the data that may influence certainty. Possible ratings were "sufficient evidence of toxicity," "limited evidence of toxicity," "inadequate evidence of toxicity," or "evidence of lack of toxicity" (Table 1), based on categories used by the International Agency for Research on Cancer (IARC), the U.S. Preventive Services Task Force, and U.S. EPA [22–25]. Review authors independently evaluated the quality of the evidence following directions as outlined in S4 Methods and then compared ratings as a group and recorded the consensus and rationale.

**Economic analysis.** We combined quantitative assessment of exposure-response from our systematic review with incidence rates of asthma and annual values of asthma control to estimate the monetized benefits of avoiding asthma in EPA's proposed rule on pressed wood products. We used the standard EPA approach of "willingness to pay" to calculate benefits, which measures the maximum amount of money that an individual is willing to pay to reduce the probability of an adverse health outcome assumed to be related to an environmental exposure [54].

To estimate the reduction in risk for asthma diagnosis, we used standardized risk estimates from our meta-analyses to estimate the reduction in risk per 1 ppb decrease in formaldehyde exposure. We assumed a Cox proportional hazard model so the number of reduced cases of asthma from a reduction in formaldehyde exposure is the exposed population times the baseline asthma risk times (1-exp(ln(OR)*(change in exposure)). Using the tables for annual asthma benefits from EPA's economic analysis for the proposed rule, we derived the exposure reduction for structures built new or renovated in the past eleven years. We used the change in indoor formaldehyde exposure for new and renovated homes at various ages (ranging from 0.124 to 3.390 ppb), the assumed baseline annual risk of asthma of 0.83%, used in EPA's economic analysis and the estimated number of children aged 4–17 in 2017 in each housing type from the U.S. Census Bureau, with the proportional hazard model to estimate the reduced number of asthma cases associated with the proposed rule [26]. We estimated the annual

Table 1.  Strength of evidence definitions for human evidence.

| Strength Rating | Definition |
|---|---|
| Sufficient evidence of toxicity | A positive relationship is observed between exposure and outcome where chance, bias, and confounding can be ruled out with reasonable confidence. The available evidence includes results from one or more well-designed, well-conducted studies, and the conclusion is unlikely to be strongly affected by the results of future studies.[1] |
| Limited Evidence of Toxicity | A positive relationship is observed between exposure and outcome where chance, bias, and confounding cannot be ruled out with reasonable confidence. Confidence in the relationship is constrained by such factors as: the number, size, or quality of individual studies, or inconsistency of findings across individual studies.[2] As more information becomes available, the observed effect could change, and this change may be large enough to alter the conclusion. |
| Inadequate Evidence of Toxicity | The available evidence is insufficient to assess effects of the exposure. Evidence is insufficient because of: the limited number or size of studies, low quality of individual studies, or inconsistency of findings across individual studies. More information may allow an assessment of effects. |
| Evidence of Lack of Toxicity | No relationship is observed between exposure and outcome, and chance, bias and confounding can be ruled out with reasonable confidence. The available evidence includes consistent results from more than one well-designed, well-conducted study at the full range of exposure levels that humans are known to encounter, and the conclusion is unlikely to be strongly affected by the results of future studies.[3] The conclusion is limited to the age at exposure and/or other conditions and levels of exposure studied. |

[1] The Navigation Guide rates the quality and strength of evidence of human and non-human evidence streams separately as "sufficient", "limited", "inadequate" or "evidence of lack of toxicity" and then these two ratings are combined to produce one of five possible statements about the overall strength of the evidence of a chemical's reproductive/developmental toxicity. The methodology is adapted from the criteria used by the International Agency for Research on Cancer (IARC) to categorize the carcinogenicity of substances except as noted.

[2] Language for the definitions of the rating categories were adapted from descriptions of levels of certainty provided by the U.S. Preventive Services Task Force Levels of Certainty Regarding Net Benefit. http://www.uspreventiveservicestaskforce.org/uspstf07/methods/benefit.htm.

[3] Language for the definitions of the rating categories were adapted from descriptions of levels of certainty provided by the U.S. Preventive Services Task Force Levels of Certainty Regarding Net Benefit.

benefits for lowering formaldehyde emissions once the impacts of the reduction have reached steady-state [26].

To quantify the economic benefits of the reduction in asthma risk, we used estimates reported in the literature for the annual willingness to pay for full asthma control (inflated to 2018 dollars) from three studies. Full asthma control is equivalent to avoiding a case of asthma. Blomquist et al. [27] used a two-stage contingent valuation survey of parents of asthmatic children aged 4–17 years and of adults to elicit the willingness to pay for a hypothetical drug that would control asthma symptoms. The mean annual willingness to pay for children was $3,434 and the mean annual value for adults was $2,368. Blumenschein and Johannesson [28] used a contingent valuation bidding game to estimate asthma patients' willingness to buy a new treatment that cured their asthma, finding a mean value of $3,621. O'Conor and Blomquist [29] used a two-stage contingent valuation survey of adults with asthma to elicit the tradeoff between hypothetical medication of varying degrees of safety and efficacy and estimated a mean annual willingness to pay for full asthma control of $2,413 using the value of statistical life. The average annual value of asthma control for adults across all three studies is $2,801 and the annual value for children is $3,434 from Blomquist et al. [27]. The total value to an individual to not develop asthma at a given age is the present discounted value (3% discount rate) of the annual values over the life expectancy of that individual.

## PRISMA Diagram of included/excluded studies

**Fig 2. PRISMA flowchart showing the literature search and screening process for studies relevant to formaldehyde exposure and asthma outcomes.** Our search was not limited by language or publication date (search was conducted up until April 1, 2020). The search terms used for each database are provided in S1–S7 Tables.

## Results

### Included studies

We retrieved a total of 4,821 unique records (4,482 from the initial search on March 15, 2016, an additional 254 from an updated search on March 15, 2018, and an additional 85 from an updated search on April 1, 2020), of which 150 ultimately met the inclusion criteria. Given the large number of diverse references identified, we decided to focus on studies where the asthma status of all study participants was measured (90 studies) (Fig 2). Our rationale was that these studies provided the most robust evidence for understanding the relationship between formaldehyde exposure and asthma because they all had quantitative measures of formaldehyde exposure, participants for whom asthma status was known, and included asthmatics. Lists of all other studies are provided in the supplemental materials (S1 Results). Several included studies contained information from multiple records, such as a graduate thesis and a published manuscript following the cohort over time; the information from these records were combined into one record and listed as the main published manuscript. Four studies were identified that

looked at similar outcomes from the same study population, so we combined these and focused on the publication for which the most relevant information was reported, supplementing with additional information from the related publications when necessary. We contacted corresponding study authors for 21 studies to request additional information missing from their published articles and received useable data from three.

Studies were further categorized separately into four combinations of study population and outcome (with some studies reporting on multiple populations/outcomes falling in multiple categories): 1) Child asthma diagnosis (n = 24); 2) Child asthma exacerbation and symptoms (n = 23); 3) Adult (general population and occupational) asthma diagnosis (n = 20); Adult (general population and occupational) asthma exacerbation and symptoms (n = 26). Presentation of results below include separate discussions for each of these four population/outcome categories. In particular, S99 Table presents study characteristics for included studies stratified by these group population/outcome categories.

**Characteristics of included studies—Demographics.** The 90 included studies were published between 1969 and 2019, were conducted in 23 different countries (including 32% (n = 29) within the U.S.), and included a range of 7 to 15,837 participants (Table 2, S99–S101 Tables).

Child studies were published relatively recently (1990–2016 for asthma diagnosis, 1984–2019 for asthma symptoms) whereas adult studies had a wider range of publication years including more older studies (Table 3). Almost half of child studies (11/24 for asthma diagnosis and 9/23 for asthma symptoms) had sample sizes greater than 1,000, whereas more adult studies had smaller sample sizes (13/20 for asthma diagnosis and 21/26 for asthma symptoms with sample size <500) (Table 3). Combined, child studies reported on a total of over 34,000 participants for asthma diagnosis and 32,000 participants for asthma symptoms. Adult studies reported on a total of over 8,000 participants for asthma diagnosis and 12,000 for asthma symptoms (S100 and S101 Tables).

A little over half (51%, n = 46) of the included studies were cross-sectional in study design, and the remainder were cohort (n = 17), controlled trials (n = 11), case-control (n = 7), case reports (n = 4), or of mixed study design (e.g., cross-sectional and case-control) (n = 5) (Table 2). A similar trend in study design was observed in that the majority of studies in all four population/outcome combinations were of cross-sectional study design. Children studies reporting on asthma diagnosis were mostly cross-sectional (58%) and case-control (21%) whereas those reporting on asthma symptoms were mostly cross-sectional (52%) and prospective cohort (22%) (Table 3). Adult studies reporting on asthma diagnosis were mostly cross-sectional (80%) and cohort (15%), and similarly for those reporting on asthma symptoms (58% cross-sectional, 27% cohort) (Table 3).

**Characteristics of included studies—Exposure measures.** Most studies (91%, n = 82) reported association estimates between asthma outcomes and quantitative measurements of formaldehyde exposure. In the remainder of studies (n = 8), although quantitative formaldehyde exposure measures were reported (leading to the study's inclusion), these estimates were not used by study authors directly to calculate association estimates, but rather they used categorized formaldehyde levels (i.e., high, medium, and low exposures) (Table 2). Formaldehyde levels were measured in school (n = 14), home (n = 30), work (n = 16), vehicles (n = 1), and outdoor environments (n = 6), as well as using personal monitors (n = 13) or given as experiment doses to healthy volunteers (n = 12) (S100 and S101 Tables). School formaldehyde measurements were used in 10 child asthma diagnosis and 10 child asthma symptom studies (and in no adult studies). Home formaldehyde measurements were used in 9 studies each for child asthma diagnosis and symptom studies and 7 studies each for adult asthma diagnosis and symptom studies. Work formaldehyde measurements were used in 6 adult asthma diagnosis

**Table 2. Summary of included studies (n = 90).**

| Study Characteristics | N (%) | Study Characteristics | N (%) |
|---|---|---|---|
| **Publication Year** | | **Formaldehyde Exposure** | |
| *1969* | 1 (1%) | *Measured exposure level* | 82 (91%) |
| *1977* | 1 (1%) | *Categorized exposure level* | 8 (9%) |
| *1980–1989* | 17 (19%) | | |
| *1990–1999* | 16 (18%) | | |
| *2000–2009* | 22 (24%) | | |
| *2010–2019* | 33 (37%) | | |
| **Study Design** | | **Study Participants*** | |
| *Case-control* | 7 (8%) | **Child** | 37 (41%) |
| *Nested case-control* | 3 (3%) | *Asthma\*\*\** | 24 (65%) |
| *Prospective cohort* | 15 (17%) | *Asthma symptoms\*\*\** | 23 (62%) |
| *Cohort* | 2 (2%) | *Pulmonary function\*\*\** | 5 (14%) |
| *Cross-sectional* | 46 (51%) | **Adult (General and occupational)** | 54 (60%) |
| *Cross-sectional and case-control* | 2 (2%) | *Asthma\*\*\** | 20 (37%) |
| *Non-randomized controlled trial* | 6 (7%) | *Asthma symptoms\*\*\** | 26 (48%) |
| *Randomized controlled trial* | 5 (6%) | *Pulmonary function\*\*\** | 35 (65%) |
| *Case report* | 4 (4%) | **Mixed child and adults** | 2 (2%) |
| **Sample Size** | | *Asthma\*\*\** | 1 (50%) |
| *0–50* | 24 (26%) | *Asthma symptoms\*\*\** | 2 (100%) |
| *51–100* | 16 (18%) | *Pulmonary function\*\*\** | 1 (50%) |
| *101–200* | 12 (13%) | **Unspecified** | 1 (1%) |
| *201–500* | 14 (16%) | *Asthma symptoms\*\*\** | 1 (100%) |
| *501–1000* | 5 (6%) | *Pulmonary function\*\*\** | 1 (100%) |
| *>1000* | 17 (19%) | | |
| *Not reported* | 2 (2%) | | |
| **Country**\*\* | | **Population Source** | |
| *Egypt, Estonia, Indonesia, Iran, Japan, Malta, New Zealand, Poland, Russia, Thailand, United Arab Emirates* | 1 (12%) | **General population (Adult and child)** | 59 (66%) |
| *Canada, Finland, Portugal, Romania* | 2 (9%) | *Asthma\*\*\** | 33 (56%) |
| *Denmark* | 3 (3%) | *Asthma symptoms\*\*\** | 30 (51%) |
| *France* | 4 (4%) | *Pulmonary function\*\*\** | 18 (31%) |

*(Continued)*

**Table 2.** (Continued)

| Study Characteristics | N (%) | Study Characteristics | N (%) |
|---|---|---|---|
| *Australia, China* | 5 (11%) | **Occupational** | 31 (34%) |
| *United Kingdom* | 5 (6%) | *Asthma*\*\*\* | 11 (35%) |
| *South Korea* | 7 (8%) | *Asthma symptoms*\*\*\* | 19 (61%) |
| *Sweden* | 13 (14%) | *Pulmonary function*\*\*\* | 20 (65%) |
| *United States* | 29 (32%) | | |

\*Studies that reported child versus adult data separately fell into both categories (as opposed to studies that reported collectively on children and adults mixed in the study population)—therefore total % is greater than 100%.

\*\*Due to the variety of different countries represented, countries with similar counts have been grouped together for reporting. For instance, there are 5 studies located in Australia and 5 other studies located in China.

\*\*\*Many studies report multiple asthma outcomes—therefore total % is greater than 100%. Percentages are calculated out of the category sub-total; for instance, the percentage of asthma studies in children is calculated as 24/37.

**Table 3. Study characteristics, stratified by population health outcome group.**

| | Child asthma n (%) | Child asthma symptoms n (%) | Adult asthma n (%) | Adult asthma symptoms n (%) |
|---|---|---|---|---|
| **Publication Year** | | | | |
| *1969* | 0 | 0 | 0 | 1 (4%) |
| *1977* | 0 | 0 | 0 | 1 (4%) |
| *1980–1989* | 0 | 1 (4%) | 2 (10%) | 6 (23%) |
| *1990–1999* | 3 (13%) | 1 (4%) | 4 (20%) | 7 (27%) |
| *2000–2009* | 7 (29%) | 6 (26%) | 5 (25%) | 6 (23%) |
| *2010–2019* | 14 (58%) | 15 (65%) | 9 (45%) | 5 (19%) |
| **Study design** | | | | |
| *Case-control* | 5 (21%) | 2 (9%) | 1 (5%) | 0 |
| *Nested case-control* | 2 (8%) | 0 | 0 | 1 (4%) |
| *Prospective cohort* | 2 (8%) | 5 (22%) | 2 (10%) | 7 (27%) |
| *Cohort* | 0 | 0 | 1 (5%) | 0 |
| *Cross-sectional* | 14 (58%) | 12 (52%) | 16 (80%) | 15 (58%) |
| *Cross-sectional and case-control* | 1 (4%) | 2 (9%) | 0 | 0 |
| *Non-randomized controlled trial* | 0 | 1 (4%) | 0 | 3 (11%) |
| *Randomized controlled trial* | 0 | 1 (4%) | 0 | 0 |
| *Case report* | 0 | 0 | 0 | 0 |
| **Sample size** | | | | |
| *0–50* | 0 | 2 (9%) | 1 (5%) | 6 (23%) |
| *51–100* | 3 (12%) | 2 (9%) | 5 (25%) | 6 (23%) |
| *101–200* | 6 (25%) | 2 (9%) | 1 (5%) | 4 (15%) |
| *201–500* | 1 (4%) | 7 (30%) | 6 (30%) | 5 (19%) |
| *501–1000* | 2 (8%) | 0 | 4 (20%) | 2 (8%) |
| *>1000* | 11 (46%) | 9 (39%) | 2 (10%) | 3 (11%) |
| *Not reported* | 1 (4%) | 1 (4%) | 1 (5%) | 0 |

studies and 11 adult symptom studies (and in no child studies). Outdoor exposure measurements were mostly used in child studies (3 studies of child asthma diagnosis, 4 for child asthma symptoms, and 2 for adult asthma diagnosis) whereas personal monitor measurements were mostly used in adult studies (5 studies of adult asthma diagnosis, 7 for adult asthma symptoms, and 2 each for child asthma diagnosis and asthma symptoms) (S100 and S101 Tables).

**Characteristics of included studies—Outcome measures.** Of the 90 total included studies, 41 evaluated asthma diagnosis outcomes (21 studies in children, 17 in adults, and 3 in both children and adults) and 48 evaluated asthma-related symptoms (22 studies in children, 25 in adults, and 1 in both children and adults). Asthma diagnosis was ascertained either by questionnaire (for instance, the International Study of asthma and Allergies in Childhood (ISAAC) [30]) medical records, or a physical examination (S100 and S101 Tables).

Studies reported on a wide range of asthma-related outcomes, including current/ever asthma (n = 33), asthma attacks (n = 3), respiratory symptoms (n = 9), wheeze (n = 32), shortness of breath/dyspnea/breathlessness (n = 17), chest tightness and pain (n = 10), pulmonary bronchial hyperresponsiveness (n = 1), asthma medication use (n = 6), hospitalizations (n = 2), emergency room visits (n = 1), and results from asthma control (n = 2), pulmonary function (n = 35), and bronchial provocation tests (n = 5) (S100 and S101 Tables).

Studies reporting on child asthma symptoms reported most commonly on wheeze (n = 16) and current/ever asthma (n = 14); all other asthma-related outcomes listed were reported in ≤5 studies. No child studies reported on outcomes of chest tightness and pain, pulmonary bronchial hyperresponsiveness, or bronchial provocation (S100 and S101 Tables).

Studies reporting on adult asthma symptoms reported most commonly on pulmonary function (n = 28), current/ever asthma (n = 19), wheeze (n = 15), and shortness of breath /dyspnea/breathlessness (n = 13), chest tightness and pain (n = 9), and respiratory symptoms (n = 6); all other asthma-related outcomes listed were reported in <5 studies. No adult studies reported on hospitalizations or emergency room visits (S100 and S101 Tables).

## Risk of bias assessment

We rated risk of bias separately by outcome (asthma diagnosis versus symptoms exacerbation), but since our ratings were ultimately identical by outcome, risk of bias results are presented by study only. A limited number (n = 3) of studies [31–33] reported results for mixed children/adult populations (aged 6–63 years); we excluded these studies from rating the quality of the evidence due to concerns with combining outcomes across a wide age range, given the unique issues in diagnosing and assessing asthma in children (especially at very young ages) compared to adults [34,35]. Overall, the majority of studies were rated "low" or "probably low" risk of bias across all domains (Fig 3, S1–S3 Figs). We evaluated the risk of bias separately by each of the four-study population/health outcome groups.

## Group 1: Childhood asthma diagnosis

Overall, the majority of childhood asthma diagnosis studies were rated "low" or "probably low" risk of bias across all domains (Fig 4). Several domains were predominantly rated "low" and "probably low" but included a small number of "probably high" ratings—source population (three "probably high" ratings), outcome assessment (four), incomplete outcome data (one), and exposure assessment (three). These were not consistent across any one study—i.e., only no study was rated "probably high" across all three of these domains. Generally, studies rated "probably high" were for similar reasons—i.e., for source population, three studies [36–38] reported high non-participation rates but failed to compare characteristics from study participants to those refusing to participate to explore potential selection bias. Similarly, for

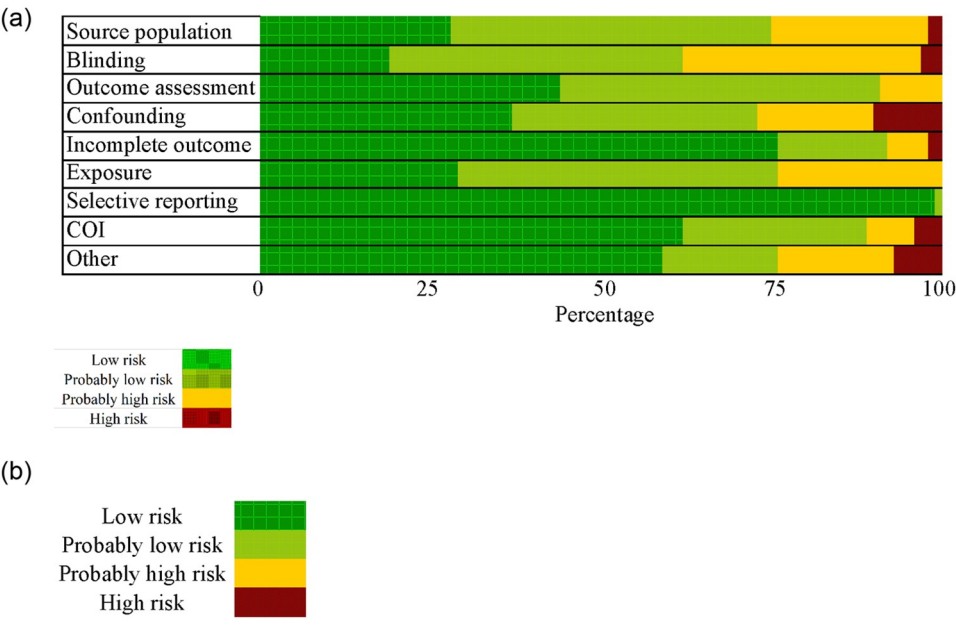

**Fig 3. Cumulative risk of bias ratings (low, probably low, probably high, or high) across all human studies included in our systematic review of formaldehyde exposure and asthma outcomes.** Risk of bias designations for individual studies are assigned by review authors according to criteria provided in S3 Methods (Risk of Bias instructions) and the justifications for each study are provided in S8–S95 Tables.

outcome assessment four studies [39–42] relied on self-reported outcomes by study participants (i.e., through a survey, self-administered spirometry, or daily diaries) but lacked follow-up by study investigators to evaluate the validity of reported outcomes. Furthermore, two studies were rated "high" risk of bias for the other category—Huang et al. [43] due to cases having formaldehyde levels sampled more during the summer when formaldehyde exposures were lower versus controls who were sampled more during the summer when formaldehyde exposures were higher and Madureira et al. [44] who published a similar paper in a different journal the year prior with similar reported results.

The most problematic domain appeared to be confounding, where six studies were rated "probably high" and four were rated as "high." Consistent with the instructions from our protocol, studies were rated as "probably high" for the confounding domain if studies evaluated some but not all of confounders pre-determined to be important (age, smoking status or exposure to environmental tobacco smoke, and socioeconomic status or parental education) and some but not all of other confounders pre-determined to be potentially important (race/ethnicity, sex, height, weight, BMI, obesity status, parental or family history of asthma, allergies, and additional environmental exposures), and were rated "high" if the study did not account for or evaluate many of the important or potentially important confounders. Studies most commonly adjusted for age, sex, and exposure to smoking. Adjusting for socioeconomic status was often accomplished through incorporating variables of family income or parent's academic background. Few studies adjusted for environmental co-exposures; those that did included exposures to allergens (house dust mites or pets), indoor dampness or mold, proximity to traffic, or certain contaminants such as nitrogen dioxide or particulate matter.

Overall, review authors felt confident that the majority of children asthma diagnosis studies were rated predominantly "low" or "probably low" risk of bias, particularly for studies that

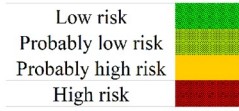

| Population/Outcome | Study | Source population | Blinding | Outcome assessment | Confounding | Incomplete outcome | Exposure assessment | Selective reporting | COI | Other |
|---|---|---|---|---|---|---|---|---|---|---|
| Children/asthma diagnosis | Zhai et al. 2013 | | | | | | | | | |
| | Madureira et al. 2015 | | | | | | | | | |
| | Zhao et al. 2008* | | | | | | | | | |
| | Annesi Maesano et al. 2012 | | | | | | | | | |
| | Yeatts et al. 2012 | | | | | | | | | |
| | Yoon and Lin 2014 | | | | | | | | | |
| | Hwang et al. 2011 | | | | | | | | | |
| | Chatzidiakou et al. 2014 | | | | | | | | | |
| | Hulin et al. 2010* | | | | | | | | | |
| | Kim et al. 2011* | | | | | | | | | |
| | Krzyzanowski et al. 1990* | | | | | | | | | |
| | Frisk et al. 2006 | | | | | | | | | |
| | Tavernier et al. 2006 | | | | | | | | | |
| | Jeong et al. 2011 | | | | | | | | | |
| | Smedje and Norback 2000 | | | | | | | | | |
| | Garrett et al. 1999 | | | | | | | | | |
| | Tuthill 1984 | | | | | | | | | |
| | Smedje and Norback 2001* | | | | | | | | | |
| | Idavain et al. 2019 | | | | | | | | | |
| | Mi et al. 2006* | | | | | | | | | |
| | Kim et al. 2007* | | | | | | | | | |
| | Hsu et al. 2012 | | | | | | | | | |
| | Huang et al. 2016 | | | | | | | | | |
| | Madureira et al. 2016 | | | | | | | | | |
| | Choi et al. 2009 | | | | | | | | | |
| | Smedje et al. 1997* | | | | | | | | | |
| Children/asthma symptoms | Zhai et al. 2013 | | | | | | | | | |
| | Delfino et al. 2003 | | | | | | | | | |
| | Yeatts et al. 2012 | | | | | | | | | |
| | Madureira et al. 2015 | | | | | | | | | |
| | Lajoie et al. 2015 | | | | | | | | | |
| | Zhao et al. 2008* | | | | | | | | | |
| | Mi et al. 2006* | | | | | | | | | |
| | Neamtiu et al. 2019 | | | | | | | | | |
| | Kim et al. 2011* | | | | | | | | | |
| | Marks et al. 2010 | | | | | | | | | |
| | Venn et al. 2003 | | | | | | | | | |
| | Jeong et al. 2011 | | | | | | | | | |
| | Dannemiller et al. 2013 | | | | | | | | | |
| | Garrett et al. 1999 | | | | | | | | | |
| | Smedje and Norback 2000 | | | | | | | | | |
| | Yon et al. 2019 | | | | | | | | | |
| | Idavain et al. 2019* | | | | | | | | | |
| | Willis et al. 2018 | | | | | | | | | |
| | Kim et al. 2007* | | | | | | | | | |
| | Rumchev et al. 2002 | | | | | | | | | |
| | Raaschou-Nielsen et al. 2010 | | | | | | | | | |
| | Fsadni et al. 2018 | | | | | | | | | |
| Children/pulmonary measures | Lajoie et al. 2015 | | | | | | | | | |
| | Madureira et al. 2015 | | | | | | | | | |
| | Krzyzanowski et al. 1990 | | | | | | | | | |
| | Marks et al. 2010 | | | | | | | | | |
| | Quackenboss et al. 1989 | | | | | | | | | |
| | Fsadni et al. 2018 | | | | | | | | | |
| Adult/asthma diagnosis | Zhai et al. 2013 | | | | | | | | | |
| | Yeatts et al. 2012 | | | | | | | | | |
| | Billionnet et al. 2011 | | | | | | | | | |
| | Wieslander et al. 1997 | | | | | | | | | |
| | Matsunaga et al. 2007 | | | | | | | | | |
| | Kilburn, Seidman, and Warshaw 1985 | | | | | | | | | |
| | Herbert et al. 1994 | | | | | | | | | |
| | Jacobsen et al. 2009 | | | | | | | | | |
| | Kilburn et al. 1985 | | | | | | | | | |
| | Elshaer et al. 2017 | | | | | | | | | |
| | Pourmahabadian et al. 2006 | | | | | | | | | |
| Adult/asthma symptoms | Zhai et al. 2013 | | | | | | | | | |
| | Yeatts et al. 2012 | | | | | | | | | |
| | Uba et al. 1989 | | | | | | | | | |
| | Wieslander et al. 1997 | | | | | | | | | |
| | Frisk et al. 2006 | | | | | | | | | |
| | Kilburn, Seidman, and Warshaw 1985 | | | | | | | | | |
| | Herbert et al. 1994 | | | | | | | | | |
| | Jacobsen et al. 2009 | | | | | | | | | |
| | Kilburn et al. 1985 | | | | | | | | | |
| | De Vos et al. 2009 | | | | | | | | | |
| | Thetkathuek et al. 2016 | | | | | | | | | |
| | Sauder et al. 1987 | | | | | | | | | |
| Adult/pulmonary measures | Quackenboss et al. 1989 | | | | | | | | | |
| | Witek, Jr et al. 1986 | | | | | | | | | |
| | Kilburn et al. 1985 | | | | | | | | | |
| | Sheppard et al. 1984 | | | | | | | | | |
| | Witek, Jr et al. 1987 | | | | | | | | | |
| | Sauder et al. 1987 | | | | | | | | | |

*Studies included in meta-analysis

Low risk
Probably low risk
Probably high risk
High risk

**Fig 4. Risk of bias ratings (low, probably low, probably high, or high) for all human studies included in our systematic review of formaldehyde exposure and asthma outcomes, organized by study population (children or adult) and outcome (asthma diagnosis, asthma symptoms, or pulmonary measures).** Risk of bias designations for individual studies are assigned by review authors according to criteria provided in S3 Methods (Risk of Bias instructions) and the justifications for each study are provided in S8–S95 Tables.

were ultimately included in the meta-analysis. In particular, of the nine studies that were ulti-mately included in the meta-analysis, four received "low" or "probably low" ratings across all risk of bias domains and accounted for 44% of the weight in estimating the overall association estimate. Studies generally that were rated "probably high" or "high" were not for reasons that were consistent across this body of evidence, and did not produce compelling reasons to downgrade the overall body of evidence as a result.

## Group 2: Childhood asthma exacerbation and symptoms

Overall, the majority of childhood asthma exacerbation and symptoms studies were rated "low" or "probably low" risk of bias across all domains (Fig 4). Several domains were predomi-nantly rated "low" and "probably low" but included a couple "probably high" or "high" ratings —blinding (one "probably high" rating), outcome assessment (two "probably high ratings), conflict of interest (one "probably high" and one "high" rating), and other (one "high" rating). These were not consistent across any one study—i.e., only no study was rated "probably high" or "high" across all domains. One study [45] was rated "probably high" for blinding because children and parents were recruited based on existence of airway respiratory symptoms and parents were responsible for deploying and retrieving in-home environmental samples and media as well as recording outcomes in diaries, thus making it unlikely that the reporting of outcomes was competed by someone without knowledge of exposure status. Two studies [42,46] were rated as "probably high" for outcome assessment due to lack of physician confir-mation or in-person interviews by study investigators to confirm asthma symptoms. One study [45] appeared to have a financial conflict of interest, with research grants provided from several private foundations from the pharmaceutical field (i.e., AstraZeneca). Another study [15] received a "high" rating for the other domain because of an apparent typographical error in the reporting of results that could not be confirmed by authors upon personal communication.

A few other domains included a higher number of "probably high" or "high" ratings— source population (five "probably high" ratings), confounding (five "probably high" and two "high" ratings), incomplete outcome data (two "probably high" and one "high" ratings), and exposure assessment (three "probably high" ratings). Similar to the child asthma diagnosis studies, the most problematic risk of bias domain appeared to be confounding, where several studies did not adjust for or consider several of the important or potentially important adjust-ment factors outlined in our protocol. Studies most commonly adjusted for age, sex, and expo-sure to smoking. Adjusting for socioeconomic status was often accomplished through incorporating variables of family income or parent's academic background. Few studies adjusted for environmental co-exposures; those that did included exposures to allergens (house dust mites or pets), indoor dampness or mold, proximity to traffic, or certain contami-nants such as nitrogen dioxide or particulate matter.

Overall, review authors felt confident that the majority of children asthma diagnosis studies were rated predominantly "low" or "probably low" risk of bias, particularly for studies that were ultimately included in the meta-analysis. In particular, of the five studies that were ulti-mately included in the meta-analysis, three received "low" or "probably low" ratings across all risk of bias domains and accounted for 90% of the weight in estimating the overall association estimate for wheeze and 100% of the weight for shortness of breath. In particular, a number of studies were rated consistently as "low" or "probably low" risk of bias across all domains, increasing the review authors' confidence that a sufficient body of evidence was available with minimal risk of bias to rate the overall body of evidence for this study population/health out-come group. Studies that were rated "probably high" or "high" were not for reasons that were

consistent across this body of evidence, and did not produce compelling reasons to downgrade the overall body of evidence as a result.

## Group 3: Adult population asthma diagnosis

Overall, the majority of adult asthma diagnosis studies were rated "low" or "probably low" risk of bias across all domains (Fig 4). Several domains were predominantly rated "low" and "probably low" but included a one to two "probably high" or "high" ratings—outcome assessment (one "probably high"), confounding (two "high"), and conflict of interest (one "probably high"). These studies were rated higher risk of bias for lack of validation for self-reported outcomes [47], failure to adjust for or consider several of the important or potentially important adjustment factors outlined in our protocol [47,48], or receiving funding from a private company without including a statement of the role of this company in influencing the study [49]. Unlike for included children studies, confounding did not appear as problematic for the adult studies, likely because many studies were occupational and relied on either matching participants based on baseline characteristics or were pre- and post-experimental tests that used each individual subject as their own control.

Other domains included a higher number of "probably high" or "high" ratings—blinding (five "probably high"), exposure assessment (five "probably high") and other (five "probably high"). These were not consistent across studies—only one study [50] received "probably high" ratings across four of these domains. This study [50] received high risk of bias ratings due to lacking detail on recruitment methods, failure to address blinding and the existing potential for bias if investigators knew exposure status of participants, exposure measurements that were assessed by self-administered, proctored questionnaires that ultimately used work assignment as a proxy for high versus low exposure groups, and the existence of potential healthy worker effect. Blinding was more generally problematic for adult studies compared to those in children since many were occupational studies where study participants were likely already aware of their exposure and/or outcome status, and blinding was not a possibility. For the other domain, all five studies that received "probably high" ratings were occupational studies where potential for healthy worker effect either likely existed or was likely.

Overall, review authors felt confident that the majority of adult asthma diagnosis studies were rated predominantly "low" or "probably low" risk of bias. In particular, one study [51] received "low" risk of bias ratings across all domains, another study [33] was rated consistently as "low" or "probably low" risk of bias across all domains, and several studies [49,52,53] only received a "probably high" rating in one category, increasing the review author's confidence that a sufficient body of evidence was available with minimal risk of bias to rate the overall body of evidence for this study population/health outcome group. Studies that were rated "probably high" or "high" were not for reasons that were consistent across this body of evidence, and did not produce compelling reasons to downgrade the overall body of evidence as a result.

## Group 4: Adult population asthma symptoms

Overall, the majority of adult asthma diagnosis studies were rated "low" or "probably low" risk of bias across all domains (Fig 4). Several domains were predominantly rated "low" and "probably low" but included one to two "probably high" or "high" ratings—source population (one "probably high" and one "high"), confounding (two "probably high"), incomplete outcome data (one "high"), exposure assessment (two "probably high"), and conflict of interest (one "probably high"). These studies were rated higher risk of bias for lacking details regarding recruiting and inclusion/exclusion criteria [50,54], failure to adjust for or consider several of

the important or potentially important adjustment factors outlined in our protocol [55,56], measureing exposure only for a portion of study participants [57], relying on self-reported outcomes by study participants but lacking follow-up for validation [50], or receiving funding from a private company without including a statement of the role of this company in influencing the study [49]. Unlike for included children studies, confounding did not appear as problematic for the adult studies, likely because many studies were occupational and relied on either matching participants based on baseline characteristics or were pre- and post-experimental tests that used each individual subject as their own control.

A few other domains included a higher number of "probably high" or "high" ratings—blinding (five "probably high" and one "high") and other (four "probably high" and one "other"). Similar to adult asthma diagnosis studies, blinding was generally more problematic for included occupational studies where study participants likely were already aware of their exposure and/or outcome status and blinding was not a possibility. For the other risk of bias domain, all five studies that received high risk of bias ratings were occupational studies where potential for healthy worker effect either likely existed or was likely (for instance, de Vos et al. [58] specifically excluded individuals with "unstable asthma, current acute or chronic respiratory illness, or any other chronic or severe illnesses," thus likely leading to selection bias that favored healthier individuals).

Overall, review authors felt confident that the majority of adult asthma diagnosis studies were rated predominantly "low" or "probably low" risk of bias. In particular, one study [51] received "low" risk of bias ratings across all domains, another study [33] was rated consistently as "low" or "probably low" risk of bias across all domains, and a number of studies [49,56,59] only received a "probably high" rating in one category, increasing the review author's confidence that a sufficient body of evidence was available with minimal risk of bias to rate the overall body of evidence for this study population/health outcome group. Studies that were rated "probably high" or "high" were not for reasons that were consistent across this body of evidence, and did not produce compelling reasons to downgrade the overall body of evidence as a result.

All adult studies with pulmonary measure outcomes received "probably high" or "high" ratings for the source population domain, each for slightly different reasons but all stemming from the fact that these were randomized controlled exposure trials with small sample sizes. For instance, Witek et al. [60] received a "probably high" rating because all 14 participants were a self-selected group of individuals responding to a recruitment advertisement (S86 Table). The 'other' risk of bias domain was used predominantly to capture healthy worker bias for included occupational studies—the phenomenon that occupations where chemical exposures occur often tend to avoid employment of older, younger, or ill individuals, and hence select out for susceptible individuals [61–63] (Figs 3 and 4). Studies considered in the meta-analysis or sensitivity analysis were generally high quality, with only "probably high" or "high" ratings in the domains blinding, outcome assessment, or confounding (Fig 4).

Occupational studies received higher risk of bias ratings for the domains of exposure assessment and 'other' compared to general population studies (S2 Fig), resulting from reliance on job exposure matrices to classify formaldehyde exposures (based solely on job titles without measuring formaldehyde levels) or potential healthy worker effects. In contrast, over a third of general population studies received "probably high" or "high" ratings for the confounding domain from failure to account for the important confounding variables as outlined in our protocol. In contrast, many occupational studies incorporated matching study participants in the study design—for example matching exposed and unexposed by age, ethnicity, or job functions from similar socioeconomic status—and thus resulted in lower risk of bias ratings for confounding.

## Statistical analysis

**Group 1: Childhood asthma diagnosis.**   Of the 37 studies reporting on child populations, 24 reported on outcomes related to asthma diagnosis (i.e., children having been diagnosed by a physician as having asthma or based on self-reported asthma diagnosis). Nine of these studies were identified as combinable in a meta-analysis [38–41,64–68]. The remaining studies could not be combined because they either categorized formaldehyde exposures or reported outcomes that could not be converted to an odds ratio (i.e., median formaldehyde exposures for those with asthma versus those without). Attempts to obtain estimates that could be standardized to an odds ratio from the study authors were unsuccessful.

One study, Rumchev et al. (2002) [15], was excluded from the meta-analysis because it included very young children (between 6 months and 3 years old), which could potentially have resulted in misclassification of infection-associated wheezing in young children as asthma [14], leading the NAS to conclude that this study should not be included in meta-analyses of formaldehyde and asthma. The estimate from another study in the meta-analysis, Krzyzanowski et al. (1990) [41] was investigated in a sensitivity analysis removing the estimate because it was the only unadjusted estimate included.

One study considered for the meta-analysis measured incident asthma cases—Smedje et al. (2001) followed children over time to identify new asthma diagnoses [40]. The remaining studies measured prevalent cases based on self-reported or physician ever having diagnosed with asthma, but because they all incorporated some requirement of current asthma symptoms (i.e., use of asthma medication or wheezing in the past 12 months) we decided that it was acceptable to combine prevalent and incident asthma cases. All studies measured indoor formaldehyde exposures, either at home or in school classrooms.

A meta-analysis combining effect estimates from the 9 children's asthma diagnosis studies using random effects modeling found an elevated OR (1.20) with 95% CI range above 1 (95% CI: [1.02, 1.41]), predicting an 20% increased odds of being diagnosed with asthma per 10-μg/m$^3$ increase in formaldehyde exposure (Fig 5). Removing the estimate from Krzyzanowski et al. [41], the only study reporting unadjusted estimates, slightly elevated the odds ratio (1.20 to 1.26) with a similar 95% CI [1.04, 1.53] (Table 4) [15].

The two most statistically influential studies in the meta-analysis were Krzyzanowski et al. [41] and Kim et al. [65]. We removed these study to determine how this might impact the overall effect estimate. The impact of removing Krzyzanowski et al. [41] as discussed above as part of the sensitivity analysis was minimally impactful; removing Kim et al. [65] had a similar null effect, only slightly elevating the odds ratio (1.27) and changing the 95% CI [1.06, 1.54] (Table 4) [68].

We used a funnel plot and used Egger's test for small-study effects to statistically test for publication bias in the eight studies in the meta-analyses. Our funnel plots revealed no evidence of overall publication bias (p-value = 0.35) (S98 Table; S4 Fig)—however, the small number of studies (<10) might result in no indication of publication bias when in fact it might exist.

We also investigated the potential impact of a new or unpublished hypothetical study necessary to alter the results of the meta-analysis. In making this calculation, we assumed that the new hypothetical study would have a standard error equal to the smallest in our group of studies—0.14 for children asthma diagnosis [41,66,68]. We determined that a new study would be required to have an estimate of OR = 0.97, 95% CI: [0.74, 1.27] to change the 95% confidence interval of the meta-analysis overlapping one. We judged the existence of a study with such a result to be possible, given that this association estimate and confidence interval was within the range of other included studies, but not likely given that this point estimate would be in the opposite direction of all studies included in the meta-analysis.

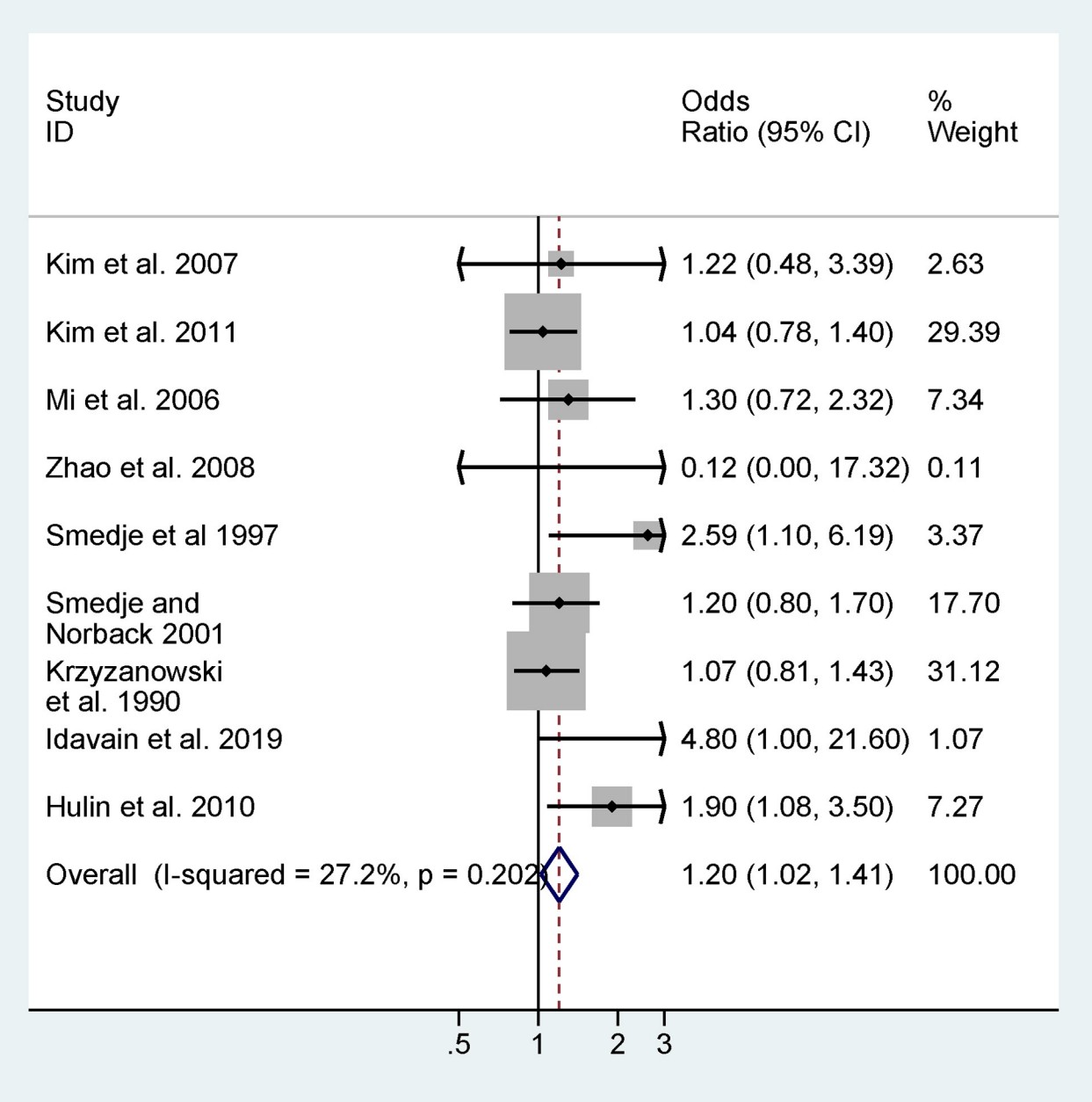

**Fig 5. Meta-analysis of human studies (n = 9 studies, including a total of 9,049 children) for formaldehyde exposure for asthma diagnosis assessed in children up to 15 years of age: Reported effect estimates and 95% confidence interval (CI) from individual studies (inverse-variance weighted, represented by size of rectangle) and overall pooled estimate from random effects (RE) model per 10 μg/m³ increase in formaldehyde exposure.** Heterogeneity statistics: $I^2$ = 27.2%, p = 0.202.

To shift our meta-analysis to have an overall association estimate just below zero (i.e., increases in formaldehyde exposures would be associated with decreases in asthma outcomes) would require a new study reporting an OR = 0.05, 95% CI: [0.04, 0.07]. We judged the existence of a well-conducted study with such a result to be very unlikely, given that this association estimate and confidence interval was considerably outside the range of the estimates from almost every included study.

**Table 4. Meta-analysis and sensitivity analysis of childhood asthma diagnosis (N = 9 studies) pooled ORs and 95% CIs for random-effects models.**

| | Number of studies | Random-effects model | |
| --- | --- | --- | --- |
| | | OR (95% CI) per 10-μg/m³ increase | I² (p-value) |
| **Asthma Diagnosis** | 9 | 1.20 (1.02, 1.41) | 27% (p = 0.2) |
| Sensitivity Analysis | | | |
| (-) Krzyzanowski et al. 1990 [41] | 8 | 1.26 (1.04, 1.53) | 31% (p = 0.18) |
| (-) Kim et al. 2011 [65] | 8 | 1.27 (1.06, 1.54) | 28% (p = 0.21) |

(-) indicates removing a study from the meta-analysis for sensitivity analysis.

Data that could not be combined into a meta-analysis were visually depicted on scatterplots when possible. The categorical odds and risk ratios (n = 14), formaldehyde levels (n = 6), and asthma prevalence (n = 5) were visually displayed for consideration in rating the overall body of evidence (S5–S7 Figs). Several studies with estimates included in the meta-analysis also reported secondary estimates (for instance, outcomes of self-reported current asthma) that were included on these scatterplots. Overall, these data appeared generally consistent with each other (i.e., increasing exposure to formaldehyde associated with increasing odds/risk ratios, asthma prevalence, and asthma status), and with the results of the meta-analysis. The secondary estimates from studies included in the meta-analysis [39,40,64–68] were also within the range of studies included in the meta-analysis (S5 Fig). Additional studies further supported the meta-analysis estimate; for instance, Tavernier et al. [36] reported odds ratios for self-reported asthma confirmed by physician by tertile of formaldehyde exposure, with an estimate of 1.22 (95%CI: [0.49, 3.07]) comparing the third to first tertile (S5 Fig). Several studies reported associations with asthma and categorical exposures to formaldehyde, which allowed review authors to evaluate the potential for a dose-response relationship. Rumchev et al. [15] reported a consistent relationship between increasing exposure (across four exposure groups ranging from 10 to >50 μg/m³) and increased odds for asthma diagnosis. However, other studies did not illustrate a similar relationship—for instance, Annesi Maesano [69] reported increased odds (OR = 1.1, 95% CI [0.87, 1.38]) for self-reported asthma comparing the medium to low tertile for formaldehyde exposure, but decreased odds (OR = 0.9, 95% CI: [0.76, 1.08]) comparing the high to low tertile (S5 Fig). Similarly, some studies reporting asthma prevalence with increasing formaldehyde exposure supported a dose-response relationship with increasing exposure [37,51,70,71] whereas others did not [41] (S6 Fig). Review authors concluded that these data supported the meta-analysis results and association between formaldehyde exposure and asthma diagnosis, but that there was limited evidence supporting a dose-response relationship.

**Group 2: Childhood asthma exacerbation and symptoms.** Twenty-three studies reported symptoms related to asthma—asthma attack, wheeze, or breathlessness/shortness of breath (Table 3). Of these, six studies [37,38,64–67] were initially identified as potentially combinable in a meta-analysis for the association between indoor formaldehyde exposures and wheeze or daytime shortness of breath. One study reported a crude OR estimate for respiratory symptoms including wheeze and shortness of breath, but did not provide an estimate of variability (i.e., confidence limits or standard error) and therefore could not be included in the meta-analysis. Efforts to contact study authors to obtain this information were unsuccessful. Thus, we ultimately combined five studies in our meta-analysis (Fig 6). Several studies provided multiple effect estimates to the meta-analysis—e.g., Kim et al. reported effect estimates for wheeze symptoms and daytime breathlessness associated with indoor formaldehyde

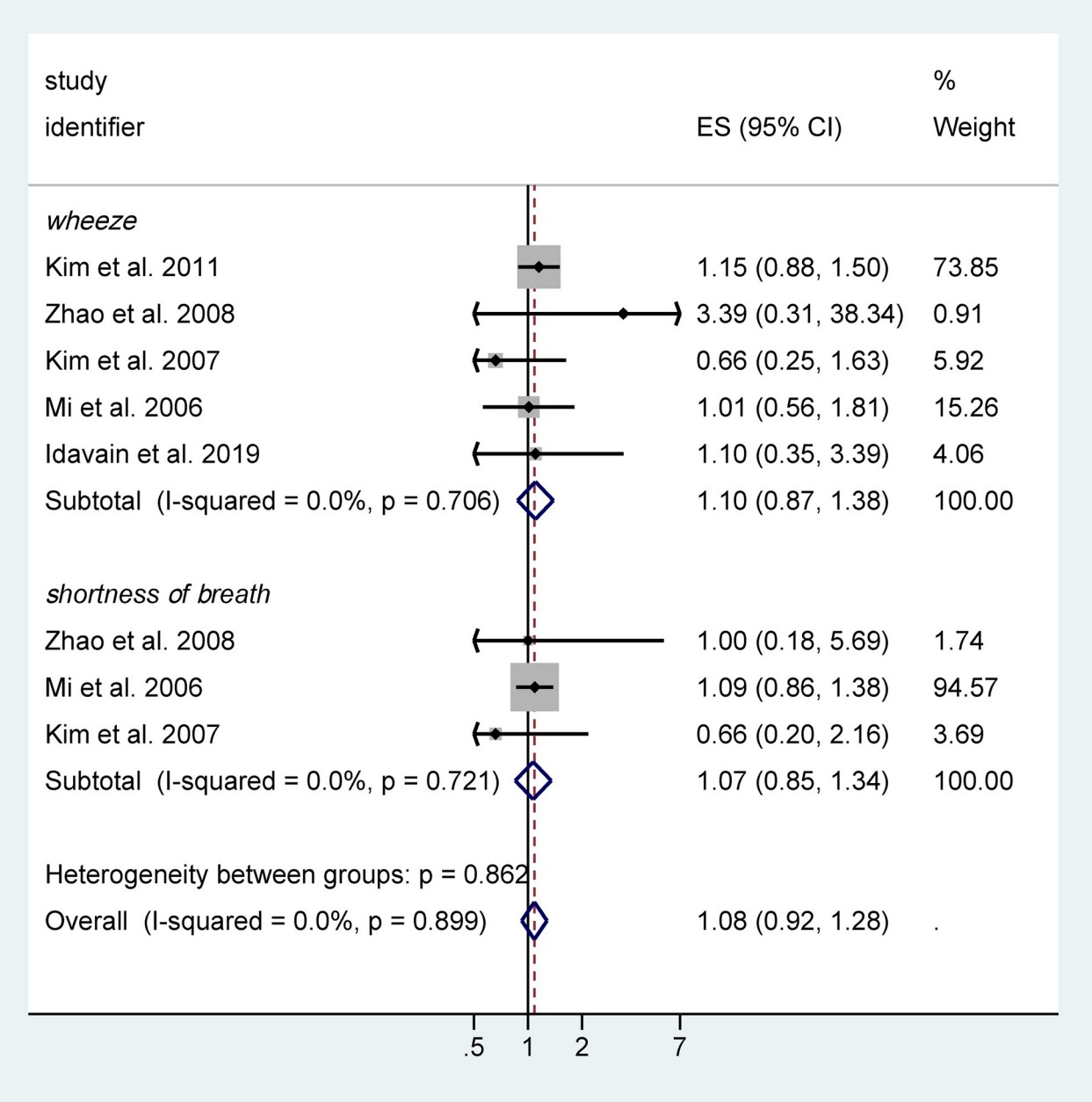

**Fig 6. Meta-analysis of human studies (n = 5 studies, including a total of 7,662 children) for formaldehyde exposure for asthma symptoms (wheeze and shortness of breath) assessed in children up to 15 years of age: Reported effect estimates and 95% confidence interval (CI) from individual studies (inverse-variance weighted, represented by size of rectangle) and overall pooled estimate from random effects (RE) model per 10 μg/m³ increase in formaldehyde exposure.** Heterogeneity statistics: $I^2$ = 0%, p = 0.899.

exposure. Overall, separate combined effects for wheeze and shortness of breath were similar and the combined effects were moderate (OR = 1.08, 95% CI: [0.92, 1.28]) (Fig 6). Due to the small number of studies contributing estimates to the meta-analysis, we did not conduct a statistical analysis of potential publication bias.

Since the meta-analysis association estimate 95% lower bound CI was below 1, we only explored the sensitivity of shifting our meta-analysis to have an overall association estimate

just below zero (i.e., such that increases in formaldehyde exposures would be associated with decreases in asthma outcomes). We assumed that the new hypothetical study would have a standard error equal to the smallest in our group of studies, 0.12 [66]. We concluded this would require a new study reporting an OR = 0.84, 95% CI: [0.66, 1.017]. We judged the existence of a well-conducted study with such a result to be possible, given that this association estimate and confidence interval was within the range and overlapped with most of the included studies and aligned with the estimate of one study in particular.

The categorical odds ratios (n = 10), formaldehyde levels by asthma status (n = 2), and symptom scores (n = 1) were visually displayed on the same figure for consideration in rating the overall body of evidence (S8 and S9 Figs). Most studies identified elevated association estimates from exposures to formaldehyde, but lower 95% CI was below 1. Several studies [38,64,65,67,72] reported on different asthma symptoms (asthma attacks, asthma symptoms, or wheeze) per 1 $\mu g/m^3$ formaldehyde exposure and reported consistent estimates of positive odds ratios ranging from 0.96–1.2 (S7 Fig). Several studies [45,73–75] reported on categorical formaldehyde exposures, but did not demonstrate a consistent dose-response relationship (S7 Fig). For instance, Raaschou-Nielsen [45] reported on wheezing symptom across five exposure categories (ranging from 0 to >25.6 $\mu g/m^3$ formaldehyde) with increased odds ratios across three groups (OR = 1.11, 1.21, 1.4) but a negative odds ratio for the highest exposure group (OR = 0.67). Review authors concluded that these data supported the meta-analysis results and association between formaldehyde exposure and asthma symptoms, but that there was limited evidence supporting a dose-response relationship.

Four studies reported pulmonary function measures in children, but because two studies reported on peak expiratory flow rates (PEFR) and two others reported on forced expiratory volume in one second ($FEV_1$) and forced vital capacity (FVC), a comparison between such a small number of studies was determined not to be useful.

**Group 3: Adult population asthma diagnosis.** Seventeen total studies included outcomes of whether subjects had been previously diagnosed by a physician with having asthma (most commonly ascertained through use of a self-reported questionnaire (n = 11) or through medical records or physician examination (n = 6)). None of these 17 studies reported sufficient data to evaluate outcomes with respect to a continuous 10-$\mu g/m^3$ increase in formaldehyde. Three studies reported results for at least two measured exposure categories; the majority of studies reported exposures categorically, such as exposed versus unexposed or by job category. Due to the small number of studies and high amount of heterogeneity in key study characteristics, the studies were not amenable to meta-analysis to combine effect estimates. We identified three studies reporting similar ranges of exposure categories to assess for a dose-response relationship for asthma diagnosis and identified a positive trend (Fig 7), although review authors noted the small number of studies and limited dose groups included.

The formaldehyde levels by categorical odds ratios (n = 4) and asthma prevalence (n = 4) were visually displayed for consideration in rating the overall body of evidence (S10 and S11 Figs). Although the categorical odds ratios varied considerably in how formaldehyde exposures were categorized (i.e., high vs. low, exposed to newly painted dwelling/workplace vs. not, occupations exposed to formaldehyde vs. not, etc.), there was a consistent increase in odds of asthma diagnosis with increased category of exposure (S10 Fig). For instance, Billionnet et al. [53] reported an increased odds (OR = 1.43) for those in the high exposure group ($\geq$28.03$\mu g/m^3$) compared to those in the low exposure group (<28.03$\mu g/m^3$). However, review authors noted a limitation with Billionnet et al. [53] in that no estimates of statistical confidence (i.e., standard error, 95% confidence interval) were reported with these estimates. Although all four studies reported increased odds with increased category of exposure, only Herbert et al. [76] reported a statistically significant increase (comparing exposed versus non-exposed

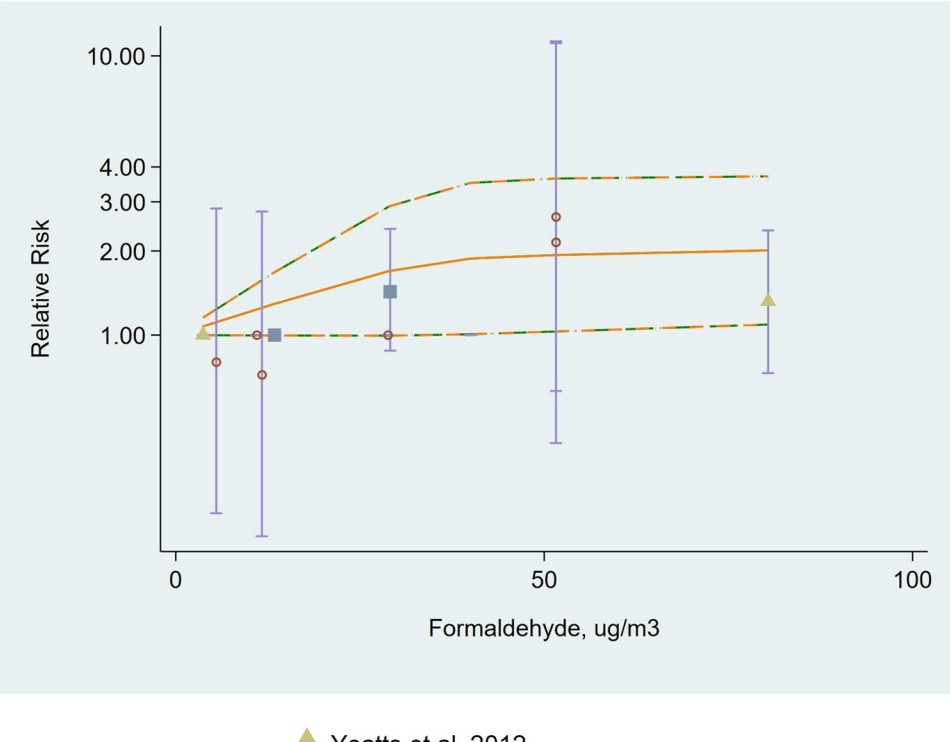

△ Yeatts et al. 2012

■ Billionnet et al. 2011

○ Matsunaga et al. 2007

**Fig 7. Dose-response relationship (n = 3 studies, including a total of 3,600 adult participants) between formaldehyde exposure (μg/m³) and relative risk of asthma diagnosis in adults.** Dose-response data from Yeatts et al. 2012 (63), Billionnet et al. 2011 (92), Matsunaga et al. 2008 (93). Data were modeled with random-effects log linear models with restricted cubic splines mixed effects methods with exchangeable covariance structure of multivariable-adjusted relative risks. Lines with long dashes represent the 95% confidence interval (CI) bounds for the fitted nonlinear trend (solid line). Symbols (triangles, circles, and squares) represent point estimates.

occupational groups). Similarly, the scatterplot of prevalence data by formaldehyde exposure categories demonstrated a similar pattern of supporting increases in asthma prevalence with increasing formaldehyde exposure (S11 Fig). For instance, Elshaer and Mahmoud [47] reported dramatic prevalence increases in exposed occupational workers for asthma (53.3%) versus non-exposed workers.

Considering the overall evidence, review authors concluded that there did appear to be evidence supporting a relationship between increasing formaldehyde exposure and asthma diagnosis, although the number of studies was low and the variety of exposure categories made it challenging to easily compare across different studies.

**Group 4: Adult population asthma symptoms.** Twenty studies reported on asthma-related symptoms—i.e., asthma attack, wheeze, or breathlessness/shortness of breath (Table 3). All studies reported categorical formaldehyde exposures and therefore could not be combined in a meta-analysis. The categorical odds ratios (n = 5), asthma prevalence (n = 4), and symptom score (n = 1) were visually displayed for consideration in rating the overall body of evidence (S9–S11 Figs). The symptom score study and most studies reporting odds ratios documented increased risk of symptoms with exposure to formaldehyde, with several

reporting statistically significant findings (S9 and S10 Figs). For instance, Herbert et al. [76] reported a statistically significant increase in asthma symptoms (attacks of wheeze) comparing exposed versus non-exposed occupational groups. Asthma prevalence estimates were generally greater with increased exposure to formaldehyde, but these studies lacked confidence intervals around the point estimates (S11 Fig). However, there were few studies reporting on prevalence outcomes and results were not consistent across studies. For instance, Kilburn, Seidman, and Warshaw [50] reported consistent increases in asthma symptom prevalence in an occupational setting with increases in the hours of exposure to formaldehyde but Thetkathuek et al. [55] reported an inconsistent relationship with wheeze symptoms across low, moderate, and high formaldehyde exposure groups (lower prevalence in the moderate exposure group compared to low exposure group).

There were also 32 total studies that reported on pulmonary lung measures in adults. We decided to focus on studies reporting associations between formaldehyde exposure and Forced Expiratory Volume in 1 second (FEV1) outcomes, following recommendations from National Institute of Health (NIH) to use FEV1 as a supplemental outcome related to asthma exacerbation. Most studies reported FEV1 outcomes (n = 27), but not all reported associations with formaldehyde exposures. Several studies reported FEV1 percentage changes comparing to baseline values (either to a comparator group or standardized values, for instance standardized predicted values based on age, height and gender published by the American Thoracic Society [77]—we decided not to plot these on the same figure due to lack of comparability across studies using different comparisons or standardized values. Of the 27 studies, 7 reported associations between FEV1 measured values with formaldehyde exposures. These were visually displayed for consideration in rating the overall body of evidence (S12 Fig). Four of the studies reported confidence intervals for association estimates that overlapped between exposed and comparator groups but did not find consistent changes in FEV1 with formaldehyde exposures (i.e., comparing formaldehyde-exposed participants to controls, two studies reported decreases in FEV1 while the other two reported increases.

Considering the overall evidence, review authors concluded that there did appear to be evidence supporting a relationship between increasing formaldehyde exposure and asthma symptoms, although the number of studies was low and the variety of exposure categories made it challenging to easily compare across different studies. Rating quality and strength of the body of evidence.

Based on the comparison of the body of evidence to pre-specified criteria in our protocol (S4 Methods), the review authors concluded that there was "moderate" quality for the body of evidence for each of the four-study population/health outcome groups (Table 5). Review authors did not apply any upgrades (for large magnitude of effect, dose-response relationship, or confounding that minimizes effect) or downgrades (for risk of bias, indirectness, inconsistency, imprecision, or publication bias) to criteria across the body of evidence, which led to the final rating of "moderate".

Review authors noted that risk of bias limitations did exist across each of the study population/health outcome groups. Concerns were generally limited to the domains of blinding, confounding, exposure assessment, and "other" (the latter being predominantly limited to occupational studies that were rated for potential healthy worker bias) domains. For instance, several child asthma diagnosis studies were rated "high" (n = 4) or "probably high" (n = 6) for confounding due to the failure to adjust for the important confounders outlined in our pre-published protocol. A number of other studies were rated as "probably high" for various other domains (source population, outcome assessment, incomplete outcome, and exposure assessment). However, review authors felt that overall a sufficient number of studies were rated

**Table 5. Summary of rating quality and strength of the human evidence, by population/outcome group.**

**A. Children asthma diagnosis**

| Category | Downgrades | Rationale |
|---|---|---|
| **Initial Rating of human evidence = Moderate** | | |
| Risk of bias | 0 | Generally risk of bias did not appear consistently problematic across all studies. The confounding domain appeared to be most frequently problematic due to failure to adjust for the important confounders outlined in the protocol; however, a number of included studies were rated as "low" or "probably low" risk of bias, including several studies ultimately included in the meta-analysis. Review authors concluded that this did not appear to warrant downgrading for risk of bias across all studies. |
| Indirectness | 0 | The population, exposure, and outcome were all directly related to the PECO statement population, exposure, and outcome. There were no concerns regarding the indirectness of evidence in supporting the study question at hand. |
| Inconsistency | 0 | Studies included in the meta-analysis have similar point estimates with overlap among the confidence intervals. Effect estimates across studies were mostly positive (showing increased risk). Estimates from the meta-analysis indicate that statistical heterogeneity was moderate, but not statistically significant ($I^2$ = 46.5%, p-value = 0.06). |
| Imprecision | 0 | No concern regarding the imprecision in effect estimates across studies. |
| Publication bias | 0 | Could not rule out publication bias, but there is no affirmative evidence of its existence—in particular, funnel plots revealed no evidence of overall publication bias (p-value = 0.35). |
| | Upgrades | |
| Large magnitude of effect | 0 | The overall effect size from the meta-analysis is small but precise. Authors concluded there was not enough evidence to warrant upgrading for this domain. |
| Dose-response | 0 | Results from the meta-analysis between formaldehyde exposure and child asthma diagnosis, which assumes a linear dose-response relationship, appeared to support the existence of an association of increasing response with increased dose. However, there was limited data to statistically evaluate whether there was a dose-response relationship, primarily due to the small number of studies and the heterogeneity in reporting of effect estimates. Review authors did not believe that results from the meta-analysis were sufficient to warrant upgrading the body of evidence for evidence of a dose-response relationship. |
| Confounding minimizes effect | 0 | There was no evidence that residual confounding influenced results. |
| **Overall Quality of Evidence** | **Moderate** | Review authors did not feel that the evidence was strong enough to warrant downgrading or upgrading the overall quality rating and came to a final conclusion of "moderate" evidence. |
| **Overall Strength of Evidence** | **Sufficient** | A positive relationship is observed between exposure and outcome where chance, bias, and confounding can be ruled out with reasonable confidence. The available evidence includes results from one or more well-designed, well-conducted studies, and the conclusion is unlikely to be strongly affected by the results of future studies. |

**B. Children asthma exacerbation and symptoms**

| Category | Downgrades | Rationale |
|---|---|---|
| **Initial Rating of human evidence = "Moderate"** | | |
| Risk of bias | 0 | Generally risk of bias did not appear consistently problematic across all studies. The confounding domain appeared to be most consistently problematic due to failure to adjust for the important confounders outlined in the protocol; however, a number of included studies were rated as "low" or "probably low" risk of bias, including several studies ultimately included in the meta-analysis. Review authors concluded that this did not warrant downgrading for risk of bias across all studies. |
| Indirectness | 0 | The population, exposure, and outcome were directly relevant to the PECO statement population, exposure, and outcome. There were no concerns regarding the indirectness of evidence in supporting the study question at hand. |
| Inconsistency | 0 | Effect estimates across studies were consistent across the body of evidence, in particular as seen by the categorical odds ratios and the prevalence data visual scatterplots. |
| Imprecision | 0 | No concern regarding the imprecision in effect estimates across studies. |
| Publication bias | 0 | Number of studies included were too small (i.e., <10) for a statistical evaluation of potential publication bias. Publication bias cannot be ruled out, but there was no affirmative evidence of its existence. We conducted a comprehensive search to identify grey literature sources (i.e., conference abstracts and graduate theses) in an attempt to identify potential publication bias and did not find evidence of such (for instance, studies reporting null or negative findings in a conference abstract that lacked a subsequent publication in the peer-reviewed literature). |
| | Upgrades | |

*(Continued)*

**Table 5.** (Continued)

| | Downgrades | Rationale |
|---|---|---|
| Large magnitude of effect | 0 | Studies that found positive relationship between exposure and outcome were interpreted as a minimal magnitude of effect; insufficient evidence to upgrade for large magnitude of effect consideration. |
| Dose-response | 0 | Results from the meta-analysis between formaldehyde exposure and children asthma exacerbation and symptoms, which assumes a linear dose-response relationship, appeared to support the existence of an association of increasing response with increased dose. However, there was not enough evidence to statistically evaluate existence of a dose-response relationship, primarily due to the small number of studies and the heterogeneity in reporting of effect estimates. Review authors did not believe that results from the meta-analysis were sufficient to warrant upgrading the body of evidence for evidence of a dose-response relationship. |
| Confounding minimizes effect | 0 | There was no evidence that residual confounding influenced results. |
| **Overall Quality of Evidence** | **Moderate** | Review authors did not feel that the evidence was strong enough to warrant downgrading or upgrading the overall quality rating and came to a final conclusion of "moderate" evidence. |
| **Overall Strength of Evidence** | **Sufficient** | A positive relationship is observed between exposure and outcome where chance, bias, and confounding can be ruled out with reasonable confidence. The available evidence includes results from one or more well-designed, well-conducted studies, and the conclusion is unlikely to be strongly affected by the results of future studies. |
| **C. Adult asthma diagnosis** | | |
| Category | Downgrades | Rationale |
| **Initial Rating of human evidence = "Moderate"** | | |
| Risk of bias | 0 | Generally risk of bias did not appear consistently problematic across all studies. Most studies were rated "low" risk of bias across most domains with only one or two "probably high" ratings, with the exception of only a few studies. Occupational studies received "probably high" ratings for blinding, exposure assessment and "other" domains, but review authors did not feel this warranted a downgrade to the overall body of evidence. |
| Indirectness | 0 | The population, exposure, and outcome were directly relevant to the PECO statement population, exposure, and outcome. There were no concerns regarding the indirectness of evidence in supporting the study question at hand. |
| Inconsistency | 0 | Effect estimates across studies were generally consistent across the body of evidence; heterogeneity likely explained by the differing study designs, and data demonstrate a tendency towards increased asthma diagnosis and therefore would not warrant a downgrade for this domain. |
| Imprecision | 0 | Confidence intervals appeared to be somewhat wide, but review authors did not feel there was enough evidence to warrant downgrading for this domain. |
| Publication bias | 0 | Publication bias cannot be ruled out, but there was no affirmative evidence of its existence. We conducted a comprehensive search to identify grey literature sources (i.e., conference abstracts and graduate theses) in an attempt to identify potential publication bias and did not find evidence of such (for instance, studies reporting null or negative findings in a conference abstract that lacked a subsequent publication in the peer-reviewed literature). |
| | Upgrades | |
| Large magnitude of effect | 0 | Studies that found positive relationship between exposure and outcome were interpreted as a minimal magnitude of effect; insufficient evidence to upgrade for large magnitude of effect consideration. |
| Dose-response | 0 | Data supported the existence of a dose-response relationship, but review authors did not feel it was strong enough to warrant an upgrade for this domain. |
| Confounding minimizes effect | 0 | There was no evidence that residual confounding influenced results. |
| **Overall Quality of Evidence** | **Moderate** | Review authors did not feel that the evidence was strong enough to warrant downgrading or upgrading the overall quality rating and came to a final conclusion of "moderate" evidence. |
| **Overall Strength of Evidence** | **Sufficient** | A positive relationship is observed between exposure and outcome where chance, bias, and confounding can be ruled out with reasonable confidence. The available evidence includes results from one or more well-designed, well-conducted studies, and the conclusion is unlikely to be strongly affected by the results of future studies. |
| **D. Adult asthma exacerbation and symptoms** | | |
| Category | Downgrades | Rationale |
| **Initial Rating of human evidence = "Moderate"** | | |
| Risk of bias | 0 | Generally risk of bias did not appear problematic across all studies. Occupational studies appeared to have probably high ratings for blinding, exposure assessment and other domains, but review authors did not feel this warranted a downgrade to the overall body of evidence. |
| Indirectness | 0 | The population, exposure, and outcome were directly relevant to the PECO statement population, exposure, and outcome. There were no concerns regarding the indirectness of evidence in supporting the study question at hand. |

(*Continued*)

**Table 5.** (Continued)

| | | |
|---|---|---|
| Inconsistency | 0 | Effect estimates across studies were generally consistent across the body of evidence; heterogeneity likely explained by other factors, and data demonstrate a tendency towards increased asthma exacerbation and symptoms and therefore would not warrant a downgrade for this domain. |
| Imprecision | 0 | Confidence intervals appeared to be somewhat wide, but review authors did not feel there was enough evidence to warrant downgrading for this domain. |
| Publication bias | 0 | Publication bias cannot be ruled out, but there was no affirmative evidence of its existence. We conducted a comprehensive search to identify grey literature sources (i.e., conference abstracts and graduate theses) in an attempt to identify potential publication bias and did not find evidence of such (for instance, studies reporting null or negative findings in a conference abstract that lacked a subsequent publication in the peer-reviewed literature). |
| | Upgrades | |
| Large magnitude of effect | 0 | Some studies illustrate large impact, but this is not consistent across the studies and so review authors concluded there was insufficient evidence to upgrade for large magnitude of effect consideration. |
| Dose-response | 0 | Some data supported the existence of a dose-response relationship, but review authors did not feel it was strong enough to warrant an upgrade for this domain. |
| Confounding minimizes effect | 0 | There was no evidence that residual confounding influenced results. |
| **Overall Quality of Evidence** | **Moderate** | Review authors did not feel that the evidence was strong enough to warrant downgrading or upgrading the overall quality rating and came to a final conclusion of "moderate" evidence. |
| **Overall Strength of Evidence** | **Sufficient** | A positive relationship is observed between exposure and outcome where chance, bias, and confounding can be ruled out with reasonable confidence. The available evidence includes results from one or more well-designed, well-conducted studies, and the conclusion is unlikely to be strongly affected by the results of future studies. |

"low" or "probably low" risk of bias across all domains, in particular several studies ultimately included in the meta-analysis (i.e., [65–68]) and review authors concluded that these limitations did not rise to the level of a downgrade, in accordance with the instructions outlined in the protocol (http://www.crd.york.ac.uk/PROSPERO/; Record ID #38766, CRD 42016038766). Review authors came to similar conclusions in evaluating the risk of bias for each of the other three study population/health outcome groups. In particular, review authors noted that many of the "high" or "probably high" risk of bias ratings were assigned to a select subgroup of studies (i.e., those with issues stemming from small sample sizes or occupational studies due to healthy worker bias concerns) but the remaining included studies did not suffer from such limitations and had minimal risk of bias concerns. Review authors did not apply downgrades to the evidence for the other domains for any of the study population/health outcome groups because there lacked sufficient evidence supporting existence of indirectness, inconsistency, imprecision, or publication bias.

Review authors also did not apply any upgrade factors for any of the study population/ health outcome groups. For child asthma diagnosis and child asthma symptoms evidence, although we were able to conduct a meta-analysis that supported an association between increasing response with increased dose (based on an assumption of model linearity), there were too few studies to support the formal analysis of a dose-response relationship. Furthermore, as discussed above visual inspections of scatterplots of data not able to be combined in a meta-analysis provided mixed evidence supporting the existence of a consistent dose-response relationship. Review authors concluded that overall this evidence was not sufficient enough to warrant upgrading the evidence for dose-response relationship.

Ultimately, review authors rated the overall strength of evidence as "sufficient" for each of the four outcome groups (Table 5), based on: a) "moderate" quality of the body of evidence; b) direction of the association (i.e., consistent evidence of a positive association between formaldehyde exposure and outcomes of either asthma diagnosis or exacerbation

in symptoms, in both adults and children; c) confidence in the association with multiple well-conducted studies (i.e., several studies were prospective cohort studies that were of "low" or "probably low" risk of bias overall; and positive and/or statistically significant overall estimates of association from the combination of similar studies in a meta-analysis (Figs 5 and 6).

### Economic analysis

We valued the outcome of avoiding a case of asthma in children, as it had the strongest support from well-conducted combinable studies with minimal risk of bias concerns. We used the OR estimate of 1.20 per 10 μg/m$^3$ (95% CI: [1.02, 1.41]) (Fig 5) based on the random effects meta-analysis model for asthma diagnosis in children from indoor formaldehyde exposure.

We rescaled this OR to estimate the reduction in risk per 1 ppb decrease in formaldehyde exposure (OR of 1.02265 per 1 ppb change in formaldehyde). We estimated that EPA's proposed rule on pressed wood products would have resulted in 2,805 fewer asthma cases annually once the impacts of the reduction has reached steady-state.

We estimated a willingness to pay for a treatment that would eliminate asthma of $75,024, which translates into total economic benefits for asthma reduction from EPA's rule of approximately $210 million annually across all children in the U.S. over a 30-year period (Table 6).

**Table 6. Cases reduced and willingness to pay for a reduction in formaldehyde exposure implied by the proposed EPA rule on pressed wood products (once the impacts of the rule have reached steady-state).**

|  | Exposure reduction (ppb) | Individuals Affected | Cases avoided | Benefits with WTP = $75,024 |
|---|---|---|---|---|
| Structure age 0–1 | -3.390085 | 599,822 | 364.0 | $27,311,030 |
| Structure age 1–2 | -2.178523 | 599,822 | 237.1 | $17,787,752 |
| Structure age 2–3 | -1.408503 | 599,822 | 154.6 | $11,599,437 |
| Structure age 3–4 | -0.926590 | 599,822 | 102.3 | $7,671,854 |
| Structure age 4–5 | -0.624871 | 599,822 | 69.2 | $5,191,181 |
| Structure age 5–6 | -0.431493 | 599,822 | 47.9 | $3,592,426 |
| Structure age 6–7 | -0.306329 | 599,822 | 34.0 | $2,553,938 |
| Structure age 7–8 | -0.229512 | 599,822 | 25.5 | $1,915,142 |
| Structure age 8–9 | -0.181581 | 599,822 | 20.2 | $1,516,000 |
| Structure age 9–10 | -0.152852 | 599,822 | 17.0 | $1,276,554 |
| Structure age 10–11 | -0.133711 | 599,822 | 14.9 | $1,116,939 |
| 0–1 years post-ren. | -2.363858 | 1,306,316 | 559.1 | $41,948,116 |
| 1–2 years post-ren. | -1.525697 | 1,306,316 | 364.3 | $27,327,908 |
| 2–3 years post-ren. | -1.002335 | 1,306,316 | 240.7 | $18,058,604 |
| 3–4 years post-ren. | -0.668362 | 1,306,316 | 161.1 | $12,086,556 |
| 4–5 years post-ren. | -0.458218 | 1,306,316 | 110.7 | $8,305,820 |
| 5–6 years post-ren. | -0.323412 | 1,306,316 | 78.3 | $5,871,128 |
| 6–7 years post-ren. | -0.239982 | 1,306,316 | 58.1 | $4,360,639 |
| 7–8 years post-ren. | -0.189089 | 1,306,316 | 45.8 | $3,437,825 |
| 8–9 years post-ren. | -0.156738 | 1,306,316 | 38.0 | $2,850,684 |
| 9–10 years post-ren. | -0.133647 | 1,306,316 | 32.4 | $2,431,347 |
| 10–11 years post-ren. | -0.124415 | 1,306,316 | 30.2 | $2,263,624 |
| **Total** |  | **20,967,514** | **2,805** | **$210,474,503** |

## Discussion

We found "sufficient" evidence of an association between exposure to formaldehyde and asthma diagnosis and asthma symptoms in children and adults. The definition of "sufficient" was predefined in our protocol (Table 1). Our review had several strengths, including that we used the Navigation Guide systematic review methodology, which specifically accounted for the weaknesses identified by the NAS in the IRIS formaldehyde assessment, i.e., explicit and transparent study selection and evaluation criteria, including exclusion of a study in which asthma may have been misclassified. Moreover, our review was based only on studies where the asthma status of participants was known and which reported quantitative measures of formaldehyde exposure, and our methods accounted for several considerations of causality as part of the evaluation, specifically, our PECO statement limited included evidence based on temporality criteria and the evaluation of the strength and quality of evidence incorporated considerations of strength, consistency, and biological gradient.

We retrieved six self-identified "systematic reviews" of formaldehyde and asthma conducted between 2011 and 2015 in the literature search for our review [78–83]. Of the three reviews with findings consistent with our review, two conducted a meta-analysis of the data [78,83] and the third cited the McGwin et al. meta-analysis [82]. The three reviews which did not find compelling evidence for an association between asthma and formaldehyde exposure did not conduct a meta-analysis, and there was a wide disparity in the number and type of papers included in these reviews. Specifically, our review included 22, 17, 17, and 20 studies on child asthma diagnosis, child asthma symptoms, adult asthma diagnosis and adult asthma symptoms, respectively.

In contrast, Patelarou et al. [81] included 2 studies on formaldehyde and asthma and wheezing in children up to 5 years old; Baur et al. [80] included 8 studies on formaldehyde and asthma in occupational settings; and Nurmatov et al. [79] included 17 studies on formaldehyde and asthma etiology, 1 study on formaldehyde and asthma exacerbation, and 14 studies on asthma etiology and exacerbation (among which the authors found a positive association between formaldehyde and wheezing in young children on the basis of a "well-conducted, low-risk of bias" randomized controlled trial, which was consistent with our findings). While none of these six self-described systematic reviews fully met all of the criteria for a systematic review as specified in the Literature Review Appraisal Toolkit (http://policyfromscience.com/lrat/about-the-lra-toolkit/), the transparency of their methods allowed for better understanding the discrepant results.

In 2016, EPA published its final rule to regulating formaldehyde in pressed wood products as well as household and other finished goods. The regulations set by this final rule did not consider the benefits of preventing asthma; estimated annualized benefits (from avoided incidence of eye irritation and nasopharyngeal cancer outcomes only) ranged from $64–186 million per year. Our results show that using assumptions consistent with EPA's proposed rule [26], the final rule excluded approximately $210 million annually in total economic benefits associated with 2,805 fewer asthma cases. Furthermore, these benefits were calculated based on the willingness to pay for asthma control, and could potentially represent an underestimate of the true valuation of one's willingness to pay for avoiding an asthma diagnosis in the first place.

Formaldehyde is a high-production volume chemical ubiquitous in homes, communities, and workplaces and asthma is a prevalent and costly chronic health outcome. While our results show that the association between exposure to formaldehyde and asthma is robust, the effect estimate is relatively small, i.e., an 8% increase in children's asthma diagnosis per 10-fold increase in exposure. These findings underscore that preventing relatively "low" risks brings

"high" health benefits when exposures are ubiquitous. Our results demonstrate that benefits analyses that inform regulatory action need to account for all relevant health outcomes as to not do so could underestimate benefits.

Formaldehyde is a well-defined respiratory irritant and has been identified as a known respiratory carcinogen in humans. There are several proposed mechanisms supporting the role of formaldehyde exposure in asthma development. Formaldehyde is a small molecule with the ability to conjugate with large serum protein molecules such as albumin. This can provoke the formation of IgE antibodies, leading to degranulation of mast cells with allergic asthma response [84]. As a small molecule, formaldehyde may bind to the amino group in proteins acquiring antigenic capacities, causing immune response with the formation of specific antibodies and triggering a local mast cell response [85]. Formaldehyde is also readily absorbed into respiratory tract tissue, where it may increase T-helper cell type 2 (Th2) mediated inflammatory response and lead to cytokine mediators (3g., IL4, IL5, and IL13) release, epithelial mucous cell metaplasia, and airway recruitment of eosinophils [84]. Lastly, formaldehyde may also react with the thiol group and interfere *S*-nitrosoglutathione function, triggering an airway response [86].

Our systematic review had several limitations. First, we focused on evaluating only studies where asthma status of all study participants was measured and excluded other studies, namely studies relevant to our PECO statement but where the asthma status of participants was unknown or there were no asthmatics included, reported no quantitative measured of formaldehyde, or non-English studies. This likely would not influence our findings as studies with missing assessments for exposure and outcome are of poorer quality. We also did not independently evaluate temporality of exposure and note that included cross-sectional studies where exposures were measured concurrent to asthma outcomes may not accurately represent exposures occurring prior to asthma outcomes.

Second, while our review documented an association between formaldehyde exposure and increased childhood asthma diagnosis, symptoms and exacerbation, it did not address whether formaldehyde directly causes childhood asthma, or rather, is a trigger for childhood asthma. Asthma is a complex chronic disease that can be challenging to diagnose accurately and for which symptoms are apparent only when there is a trigger. The trigger does not necessarily cause 'asthma', but will cause an 'asthma flare up', which helps lead to the diagnosis. Thus, it is possible that formaldehyde is a 'trigger' for a child who is yet to be diagnosed with asthma or it can be that formaldehyde exposure leads to the development of asthma. It is impossible to determine this unless without a human interventional study.

Third, key estimates utilized in the economic analysis (i.e., baseline asthma risk and willingness to pay for asthma reduction) were U.S.-based estimates. Thus, the economic evaluation and monetized value of benefits from formaldehyde exposure reduction may not be directly applicable in other global settings. However, inclusion of studies in the systematic review was not limited by geographic location and we ultimately included studies from a variety of countries (Sweden, France, Australia, China, South Korea, Denmark, Finland, Poland, Portugal, United Kingdom, New Zealand, Romania, Russia, Japan, Indonesia, Thailand, Iran, the United Arab Emirates), with the first five countries in addition to the U.S. contributing to the meta-analysis estimates. Thus, results and conclusions from the systematic review are likely relevant to international settings and results from the economic analyses may be modified with geographic-specific estimates to gauge potential economic benefits in international settings.

Our results underscore that the inability to combine studies in a meta-analysis due to lack of reporting in published studies is a major challenge for systematic reviews in environmental health specifically, and for environmental health decision-making more broadly. The association between asthma and formaldehyde exposure is well-studied, as demonstrated by the large

number of epidemiology studies. However, even with a large number of included studies, there were multiple limitations to the studies that restricted our ability to combine estimates into a meta-analysis—for instance, if studies only reported categorical formaldehyde exposures or if they did not report odds ratio or relative risk estimates. Visual scatterplots of data assisted review authors' evaluation of the consistency and interpretation of data results, but many studies did not provide data amenable to extraction for scatterplots. For example, of the 26 adult (occupational and general population) asthma diagnosis studies, only 17 studies included outcome data on a physician diagnosis; none of these 17 studies reported sufficient data to evaluate outcomes with respect to a continuous increase in formaldehyde; and few studies reported results for at least two measured exposure categories. Hence, quantitative data from 9 papers were not reported in a manner that they could be objectively incorporated (i.e., not using the author's conclusions but rather just by extracting the data) into this review. Checklists such as Strengthening the Reporting of Observational Studies in Epidemiology (STROBE) guidelines for observational human studies to guide the reporting of elements necessary to describe studies comprehensively and transparently may assist with these efforts and have already been incorporated into the publication process of several high-impact journals. Furthermore, journal reviews and editors may contribute to addressing this issue by requesting increased reporting or open-access of quantitative data in a format conducive to future data analyses. Conducting a systematic review prior to the development and initiation of a new study could help design efficient studies that are intended to build on existing data and address research gaps intentionally to support future systematic reviews, risk assessment, and timely decision-making on environmental chemicals.

## Conclusion

The review authors concluded that there was "sufficient" evidence supporting an association between childhood and adult exposures to formaldehyde with asthma diagnosis and symptoms. Although studies supported modest associations (our meta-analysis for childhood exposure to formaldehyde with asthma symptoms resulted in a combined OR = 1.08), ubiquitous exposure to formaldehyde can result in potentially large impacts to population health. Our economic analysis identified annual benefits of 2,805 fewer asthma cases in the U.S.; the total economic benefit for asthma reduction from U.S. EPA's rule would be approximately $210 million annually. Thus, excluding asthma health outcomes when conducting regulatory benefit-cost analysis can underestimate the true population benefits and lead to decisions that are not fully protective of the public. Although these economic estimates are specific to the U.S., the inclusion of studies from broad geographic range indicate that results and conclusions from the systematic review are likely relevant to international settings. Our findings document that preventing formaldehyde exposure in adults and children could reduce the occurrence and impacts of a serious, chronic disease and provide significant health and economic benefits.

## Supporting information

**S1 Checklist.**
(DOC)

**S1 Fig. Risk of bias ratings for prospective cohort studies, by year.**
(DOCX)

**S2 Fig. Risk of bias rating, by study population.**
(DOCX)

**S3 Fig. Risk of bias rating, by study design.**
(DOCX)

**S4 Fig. Funnel plot for meta-analysis.**
(DOCX)

**S5 Fig. Scatterplot of categorical odds ratios not included in child asthma diagnosis meta-analysis.**
(DOCX)

**S6 Fig. Scatterplot of prevalence data not included in child asthma diagnosis meta-analysis.**
(DOCX)

**S7 Fig. Scatterplot of child and adult formaldehyde exposures by asthma status.**
(DOCX)

**S8 Fig. Scatterplot of categorical odds ratio not included in child asthma symptoms meta-analysis.**
(DOCX)

**S9 Fig. Scatterplot of child and adult symptom score not included in meta-analysis.**
(DOCX)

**S10 Fig. Scatterplot of adult categorical odds ratios.**
(DOCX)

**S11 Fig. Scatterplot of adult asthma prevalence.**
(DOCX)

**S12 Fig. Scatterplot of adult FEV1 measures.**
(DOCX)

**S1 Table. PubMed search terms.**
(DOCX)

**S2 Table. Web of science search terms.**
(DOCX)

**S3 Table. Biosis previews search terms.**
(DOCX)

**S4 Table. Embase search terms.**
(DOCX)

**S5 Table. Toxline and DART search terms.**
(DOCX)

**S6 Table. Toxicological websites/databases.**
(DOCX)

**S7 Table. Grey literature websites/databases S8–S92 Tables. Risk of bias ratings for included studies.**
(DOCX)

**S8 Table. Characteristics of Akbar Khanzadeh et al. 1994.**
(DOCX)

**S9 Table. Characteristics of Akbar Khanzadeh et al. 1997.**
(DOCX)

**S10 Table. Characteristics of Annesi Maesano et al. 2012 [69].**
(DOCX)

**S11 Table. Characteristics of Billionnet et al. 2011 [53].**
(DOCX)

**S12 Table. Characteristics of Burge et al. 1984.**
(DOCX)

**S13 Table. Characteristics of Chatzidiakou et al. 2014 [70].**
(DOCX)

**S14 Table. Characteristics of Choi et al. 2009.**
(DOCX)

**S15 Table. Characteristics of Dannemiller et al. 2013.**
(DOCX)

**S16 Table. Characteristics of De Vos et al. 2009 [58].**
(DOCX)

**S17 Table. Characteristics of Delfino et al. 2003.**
(DOCX)

**S18 Table. Characteristics of Dumas et al. 2017.**
(DOCX)

**S19 Table. Characteristics of Elshaer et al. 2017 [47].**
(DOCX)

**S20 Table. Characteristics of Ezratty et al. 2007.**
(DOCX)

**S21 Table. Characteristics of Fornander et al. 2014.**
(DOCX)

**S22 Table. Characteristics of Fransman et al. 2003.**
(DOCX)

**S23 Table. Characteristics of Frey et al. 2014.**
(DOCX)

**S24 Table. Characteristics of Frigas et al. 1984.**
(DOCX)

**S25 Table. Characteristics of Frisk et al. 2002.**
(DOCX)

**S26 Table. Characteristics of Frisk et al. 2006 [56].**
(DOCX)

**S27 Table. Characteristics of Frisk et al. 2009.**
(DOCX)

**S28 Table. Characteristics of Gannon et al. 1995.**
(DOCX)

**S29 Table. Characteristics of Garrett et al. 1999.**
(DOCX)

**S30 Table. Characteristics of Gorski et al. 1991.**
(DOCX)

**S31 Table. Characteristics of Green et al. 1987.**
(DOCX)

**S32 Table. Characteristics of Hanson et al. 1993.**
(DOCX)

**S33 Table. Characteristics of Harving et al. 1990.**
(DOCX)

**S34 Table. Characteristics of Hendrick et al. 1977.**
(DOCX)

**S35 Table. Characteristics of Herbert et al. 1988.**
(DOCX)

**S36 Table. Characteristics of Horvath et al. 1988.**
(DOCX)

**S37 Table. Characteristics of Hsu et al. 2012.**
(DOCX)

**S38 Table. Characteristics of Huang et al. 2016 [43].**
(DOCX)

**S39 Table. Characteristics of Hulin et al. 2010 [68].**
(DOCX)

**S40 Table. Characteristics of Hwang et al. 2011.**
(DOCX)

**S41 Table. Characteristics of Jacobsen et al. 2009 [57]\*.**
(DOCX)

**S42 Table. Characteristics of Jeong et al. 2011 [71].**
(DOCX)

**S43 Table. Characteristics of Kilburn et al. 1985 [50].**
(DOCX)

**S44 Table. Characteristics of Kilburn, Seidman, and Warshaw 1985 [50].**
(DOCX)

**S45 Table. Characteristics of Kim et al. 2007 [64].**
(DOCX)

**S46 Table. Characteristics of Kim et al. 2011 [65].**
(DOCX)

**S47 Table. Characteristics of Kim et al. 2014.**
(DOCX)

**S48 Table. Characteristics of Kriebel et al. 1993.**
(DOCX)

**S49 Table. Characteristics of Kriebel et al. 2001**$^*$.
(DOCX)

**S50 Table. Characteristics of Krzyzanowski et al. 1990 [41].**
(DOCX)

**S51 Table. Characteristics of Lajoie et al. 2015.**
(DOCX)

**S52 Table. Characteristics of Liu et al. 1991.**
(DOCX)

**S53 Table. Characteristics of Lofstedt et al. 2009**$^*$.
(DOCX)

**S54 Table. Characteristics of Lofstedt et al. 2011.**
(DOCX)

**S55 Table. Characteristics of Low et al. 1985.**
(DOCX)

**S56 Table. Characteristics of Madureira et al. 2015 [73].**
(DOCX)

**S57 Table. Characteristics of Madureira et al. 2016 [44].**
(DOCX)

**S58 Table. Characteristics of Malaka et al. 1990.**
(DOCX)

**S59 Table. Characteristics of Mapou et al. 2013.**
(DOCX)

**S60 Table. Characteristics of Marks et al. 2010.**
(DOCX)

**S61 Table. Characteristics of Matsunaga et al. 2007.**
(DOCX)

**S62 Table. Characteristics of Mi et al. 2006 [66].**
(DOCX)

**S63 Table. Characteristics of Milton et al. 1996.**
(DOCX)

**S64 Table. Characteristics of Norback et al. 1995.**
(DOCX)

**S65 Table. Characteristics of Norback et al. 2000 [42].**
(DOCX)

**S66 Table. Characteristics of Nordman et al. 1985.**
(DOCX)

**S67 Table. Characteristics of Popa et al. 1969.**
(DOCX)

**S68 Table. Characteristics of Pourmabahabadian et al. 2006.**
(DOCX)

**S69 Table. Characteristics of Quackenboss et al. 1989.**
(DOCX)

**S70 Table. Characteristics of Raaschou-Nielsen et al. 2010 [45].**
(DOCX)

**S71 Table. Characteristics of Rumchev et al. 2002 [15].**
(DOCX)

**S72 Table. Characteristics of Sauder et al. 1987 [54].**
(DOCX)

**S73 Table. Characteristics of Schachter et al. 1987.**
(DOCX)

**S74 Table. Characteristics of Schenker et al. 1982 [31].**
(DOCX)

**S75 Table. Characteristics of Sheppard et al. 1984.**
(DOCX)

**S76 Table. Characteristics of Smedje and Norback 2000 [42].**
(DOCX)

**S77 Table. Characteristics of Smedje and Norback 2001 [40].**
(DOCX)

**S78 Table. Characteristics of Smedje et al. 1997 [39].**
(DOCX)

**S79 Table. Characteristics of Tavernier et al. 2006 [36].**
(DOCX)

**S80 Table. Characteristics of Tuomainen et al. 2013.**
(DOCX)

**S81 Table. Characteristics of Tuthill 1984.**
(DOCX)

**S82 Table. Characteristics of Uba et al. 1989 [59].**
(DOCX)

**S83 Table. Characteristics of Venn et al. 2003 [75].**
(DOCX)

**S84 Table. Characteristics of Veremchuk et al. 2016.**
(DOCX)

**S85 Table. Characteristics of Wieslander et al. 1997 [49].**
(DOCX)

**S86 Table. Characteristics of Witek, Jr et al. 1986 [60].**
(DOCX)

**S87 Table. Characteristics of Witek, Jr et al. 1987.**
(DOCX)

**S88 Table. Characteristics of Yeatts et al. 2012 [33].**
(DOCX)

**S89 Table. Characteristics of Yoon and Lin 2014.**
(DOCX)

**S90 Table. Characteristics of Zammit-Tabona et al. 1983.**
(DOCX)

**S91 Table. Characteristics of Zhai et al. 2013 [51].**
(DOCX)

**S92 Table. Characteristics of Zhao et al. 2008 [67].**
(DOCX)

**S93 Table. Characteristics of Neamtiu et al. 2019.**
(DOCX)

**S94 Table. Characteristics of Yon et al. 2019 [72].**
(DOCX)

**S95 Table. Characteristics of Fsadni et al. 2018.**
(DOCX)

**S96 Table. Characteristics of Idavain et al. 2019 [38].**
(DOCX)

**S97 Table. Characteristics of Willis et al. 2018 [46].**
(DOCX)

**S98 Table. Egger's test for meta-analysis.**
(DOCX)

**S99 Table. Study categorization by population/outcome.**
(DOCX)

**S100 Table. Study characteristics by population/outcome.**
(DOCX)

**S101 Table. Study characteristics by study design.**
(DOCX)

**S1 Methods. Exclusion criteria for screening references.**
(DOCX)

**S2 Methods. Data Extraction fields.**
(DOCX)

**S3 Methods. Risk of Bias instructions.**
(DOCX)

**S4 Methods. Rating quality of evidence.**
(DOCX)

**S1 Results. List of included studies not considered.**
(DOCX)

**S2 Results. List of excluded studies.**
(DOCX)

**S1 References.**
(DOCX)

## Acknowledgments

ICF (Sorina Eftim (data extraction, evaluating risk of bias, initial statistical analysis); Pamela Hartman (data extraction, evaluating risk of bias); Ali Goldstone (data extraction, evaluating risk of bias, initial statistical analysis); Elizabeth Maull (data extraction, evaluating risk of bias); Dave Burch (data extraction).

Lesley Skalla, MDB Inc. (Information Specialist)

Andy Shapiro (HAWC expertise)

Brianna N. VanNoy, George Washington University (QA/QC of data extraction)

Swati Rayasam, UCSF (formatting and editing of manuscript)

We gratefully acknowledge the following corresponding study authors for providing additional information and data for our analysis:

Gitte Jacobsen

David Kriebel

Michal Krzyzanowski

Håkan Löfstedt

Gerald McGwin

Dave McClean

Karin Yeatts

## Disclaimer

The views expressed in this paper are those of the authors and do not necessarily reflect the view or policies of the U.S. Environmental Protection Agency. MDC is a member of the United States Preventative Services Task Force (USPSTF). This article does not necessarily represent the views and policies of the USPSTF.

## Author Contributions

**Conceptualization:** Juleen Lam, Patrice Sutton, Tracey J. Woodruff.

**Data curation:** Juleen Lam, Erica Koustas, Patrice Sutton, Amy M. Padula, Michael D. Cabana, Hanna Vesterinen, Natalyn Daniels, Evans Whitaker, Tracey J. Woodruff.

**Formal analysis:** Juleen Lam, Hanna Vesterinen, Charles Griffiths, Mark Dickie.

**Funding acquisition:** Juleen Lam, Patrice Sutton, Tracey J. Woodruff.

**Investigation:** Juleen Lam, Erica Koustas, Patrice Sutton, Amy M. Padula, Michael D. Cabana, Hanna Vesterinen, Charles Griffiths, Mark Dickie, Natalyn Daniels, Evans Whitaker, Tracey J. Woodruff.

**Methodology:** Juleen Lam, Erica Koustas, Patrice Sutton, Amy M. Padula, Michael D. Cabana, Hanna Vesterinen, Charles Griffiths, Mark Dickie, Natalyn Daniels, Evans Whitaker, Tracey J. Woodruff.

**Project administration:** Juleen Lam, Natalyn Daniels.

**Resources:** Juleen Lam.

**Software:** Juleen Lam.

**Supervision:** Juleen Lam.

**Validation:** Juleen Lam, Erica Koustas, Hanna Vesterinen, Natalyn Daniels.

**Visualization:** Juleen Lam, Hanna Vesterinen.

**Writing – original draft:** Juleen Lam, Charles Griffiths, Mark Dickie.

**Writing – review & editing:** Juleen Lam, Erica Koustas, Patrice Sutton, Amy M. Padula, Michael D. Cabana, Hanna Vesterinen, Charles Griffiths, Mark Dickie, Evans Whitaker, Tracey J. Woodruff.

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
