## [Decision Letter · Decision Letter 0]

16 Dec 2019

PONE-D-19-21348

Exposure to Formaldehyde and Asthma Outcomes: A Systematic Review, Meta-Analysis, and Economic Assessment

PLOS ONE

Dear Dr. Woodruff,

Thank you for submitting your manuscript to PLOS ONE. After careful consideration, we feel that it has merit but does not fully meet PLOS ONE’s publication criteria as it currently stands. Therefore, we invite you to submit a revised version of the manuscript that addresses the points raised during the review process.

We would appreciate receiving your revised manuscript by Jan 30 2020 11:59PM. To enhance the reproducibility of your results, we recommend that if applicable you deposit your laboratory protocols in protocols.io, where a protocol can be assigned its own identifier (DOI) such that it can be cited independently in the future. For instructions see: http://journals.plos.org/plosone/s/submission-guidelines#loc-laboratory-protocols

We look forward to receiving your revised manuscript.

Kind regards,

A. Kofi Amegah, PhD

Academic Editor

PLOS ONE

Additional Editor Comments:

The justification for the work is weak in light of the previous systematic reviews on the topic. Tighten up the introduction to help gauge the paper's contribution to the existing literature and its public health relevance beyond the US population.

Is there a reason why SCOPUS database was not searched?

Studies that met the included criteria as evident are those in bin1 and should be profiled as such. Subsequently, delete all text referring to the other bins in the methods, results and discussion section. The flow diagram should be revised accordingly

Did the case reports (n=4) considered in the bin1 studies satisfied the PECO guidelines in terms of the "Comparator"?

The Risk of bias assessment section is too winding and confusing. Please, simplify

Elaborate the plausible biological mechanism of the relationship in the Discussion section. Prune down the Discussion to make room for the mechanistic pathway

The conclusion should touch on the far-reaching implications of the findings (if any). It appears it has no relevance beyond the US population.

Journal Requirements:

2. Please provide the complete PROSPERO registration number in your manuscript (CRD42016038766).

3. Please provide the search start date (if searching from database inception, please specify this).

4. Please place each table in your manuscript file directly after the paragraph in which it is first cited (read order). Do not submit your tables in separate files unless they are being included as supplementary information. Tables require a label (e.g., “Table 1”) and brief descriptive title to be placed above the table. Place legends, footnotes, and other text below the table.

5. Please note that your currently uploaded Table 2 has the title “Table 3. Summary of included studies (n=85)”. Please amend as necessary.

6. Thank you for stating the following financial disclosure:

"JPB Foundation, Grant #681, https://www.jpbfoundation.org: JL, EK, PS, MDC, HV, MD, ND, EW, TJW.

NIEHS, P01ES022841, https://www.niehs.nih.gov/: JL, PS, AMP, ND, TJW

USEPA, RD 83543301, https://www.epa.gov/: JL, PS, AMP, ND, TJW"

7. Please include your tables as part of your main manuscript and remove the individual files. Please note that supplementary tables (should remain/ be uploaded) as separate "supporting information" files

9. Please upload a copy of "Supplemental Methods 5" which you refer to in your text on page 12.

10. Please upload a copy of "Supplemental Materials 3" which you refer to in your text on page 11.

Reviewers' comments:

Reviewer's Responses to Questions

**Comments to the Author**

1. Is the manuscript technically sound, and do the data support the conclusions?

Reviewer #1: Yes

Reviewer #2: Partly

2. Has the statistical analysis been performed appropriately and rigorously? 

Reviewer #1: Yes

Reviewer #2: Yes

3. Have the authors made all data underlying the findings in their manuscript fully available?

Reviewer #1: Yes

Reviewer #2: Yes

4. Is the manuscript presented in an intelligible fashion and written in standard English?

Reviewer #1: Yes

Reviewer #2: Yes

5. Review Comments to the Author

Reviewer #1: The topic of this paper is scientifically interesting and potentially relevant to a regulatory process in the United States. In prior work, some of the authors have taken thoughtful steps toward adapting popular systematic review guidelines for environmental health evaluations. While I have major reservations about the appropriateness of highly-structured review methods, like the GRADE method used here and related approaches, my comments on this paper are not founded on these objections, as such protocols are established in the literature.

Overall, this paper is technically well-executed. The authors establish explicit, state-of-the-art protocols for their systematic review and meta-analysis and follow them carefully, with full reporting of the methods and results. The findings are credible, but I have several suggestions for improving the paper, outline below.

Study selection. I doubt the decision to include very small studies and case reports. These studies are unlikely to contribute significantly to assessing causality or to quantitative analysis. Nothing in systematic review methodology requires including them as long as inclusion/exclusion criteria are explicit and applied consistently.

Data extraction. While contacting authors for unpublished data gives an appearance of thoroughness, it actually detracts from transparency and reproducibility of the results, as the information obtained would not necessarily be available to others. It is preferable for studies missing key data to be excluded.

Temporality. About half of the included studies are cross-sectional. This appears consistent with the PECO statement, which says that formaldehyde exposure can be “prior or concurrent to” any of several outcomes. However, there is an implicit assumption that current exposures are representative of those before the outcomes appeared, which may be more tenable for symptoms and exacerbation than for diagnosis. The authors should clarify how they assessed the temporality of exposure.

Writing. The paper is clearly written, but very long. There is a lot of descriptive information and some sections overlap (e.g., the descriptions of data extraction on p 8 and analysis on p 10). Readers would be grateful if the text were shortened by a third, omitting many of the descriptive details. For example, it’s obvious that only the “Bin 1” studies are informative, so there is no need for details of the others.

Risk of bias assessment. It is puzzling that many studies, notably cohort studies according to the authors, were rated “probably high risk of bias” for blinding. Blinding is exceedingly rare in observational studies of environmental exposures and knowledge of participants’ exposure status is not usually seen as an important source of bias. The authors should explain why they believe it is important to downgrade studies on this basis.

Discussion. The authors make an important point that many papers lack data needed for quantitative meta-analysis or risk assessment. However, this is not a simple problem of reporting that can be solved with checklists like STROBE. It is a problem of study conceptualization and data analysis. Journal reviewers and editors could help to address this by asking authors to report quantitative data and to do so in a form that can be used for further analysis.

Reviewer #2: I applaud the authors for identifying a public health problem (e.g., EPA proposed formaldehyde rule does not consider asthma as a health outcome) and conducting a comprehensive approach to improve public health. This approach consisted of a series of systematic reviews and associated meta-analyses, and an economic assessment on exposure to formaldehyde and asthma. I agree that excluding established health effects from the economic analysis can underestimate the benefits of regulations. However, the quality of the health hazard assessment is not adequate or transparent to support the level of evidence conclusions, and thus, opens the door for major criticism and compromises potential public impacts from the economic assessment.

Please see the attached file for details

6. PLOS authors have the option to publish the peer review history of their article (what does this mean?). If published, this will include your full peer review and any attached files.

Reviewer #1: No

Reviewer #2: No

---

## [Author Response · Author response to Decision Letter 0]

17 Jun 2020

Editor: I have incorporated all of your suggestions in to my revision. They were very helpful, thank you.

Reviewer 1: I have incorporated all of your suggestions in to my revision. They were very helpful, thank you.

Reviewer 2: I have incorporated all of your suggestions in to my revision. They were very helpful, thank you.

---

## [Decision Letter · Decision Letter 1]

28 Jul 2020

PONE-D-19-21348R1

Exposure to Formaldehyde and Asthma Outcomes: A Systematic Review, Meta-Analysis, and Economic Assessment

PLOS ONE

Dear Dr. Woodruff,

Thank you for submitting your manuscript to PLOS ONE. After careful consideration, we feel that it has merit but does not fully meet PLOS ONE’s publication criteria as it currently stands. Therefore, we invite you to submit a revised version of the manuscript that addresses the points raised during the review process.

We look forward to receiving your revised manuscript.

Kind regards,

A. Kofi Amegah, PhD

Academic Editor

PLOS ONE

Additional Editor Comments (if provided):

The organization and quality of the figure and tables poses a lot of challenges to the review process and should be improved. The font size for instance is too small and I recommend creating smaller tables to help increase the font size and enhance the presentation. A summary table with results for each assessment is also recommended. Please ensure that the responses to the reviewer's comments are systematic and come immediately after the comment to help better interrogate the revised manuscript. Further, the authors should ensure they have adequately addressed all the additional comments raised to help bring finality to the review process as quickly as possible

Reviewers' comments:

Reviewer's Responses to Questions

**Comments to the Author**

1. If the authors have adequately addressed your comments raised in a previous round of review and you feel that this manuscript is now acceptable for publication, you may indicate that here to bypass the “Comments to the Author” section, enter your conflict of interest statement in the “Confidential to Editor” section, and submit your "Accept" recommendation.

Reviewer #2: (No Response)

2. Is the manuscript technically sound, and do the data support the conclusions?

Reviewer #2: Partly

3. Has the statistical analysis been performed appropriately and rigorously? 

Reviewer #2: Yes

4. Have the authors made all data underlying the findings in their manuscript fully available?

Reviewer #2: Yes

5. Is the manuscript presented in an intelligible fashion and written in standard English?

Reviewer #2: Yes

6. Review Comments to the Author

Reviewer #2: I appreciate the authors’ efforts to improve the transparency and strengthen the scientific rigor of the systematic review. However, the revised manuscript is still not adequate to determine whether the body of evidence supports the authors’ conclusions using their systematic review method. comments and recommendations are in the attached file.

7. PLOS authors have the option to publish the peer review history of their article (what does this mean?). If published, this will include your full peer review and any attached files.

Reviewer #2: No

---

## [Author Response · Author response to Decision Letter 1]

18 Sep 2020

Please find an excel sheet with our response to reviewers in the included uploads/attachments as well as in our cover letter. Do let us know if you have trouble with this.

---

## [Decision Letter · Decision Letter 2]

13 Oct 2020

PONE-D-19-21348R2

Exposure to Formaldehyde and Asthma Outcomes: A Systematic Review, Meta-Analysis, and Economic Assessment

PLOS ONE

Dear Dr. Woodruff,

Thank you for submitting your manuscript to PLOS ONE. After careful consideration, we have decided that your manuscript does not meet our criteria for publication and must therefore be rejected.

I am sorry that we cannot be more positive on this occasion, but hope that you appreciate the reasons for this decision.

Yours sincerely,

A. Kofi Amegah, PhD

Academic Editor

PLOS ONE

Additional Editor Comments (if provided):

The manuscript suffers from poor organization and formatting of tables especially table 3 which is too voluminous. Suggestions by reviewer 2 to improve readability of the manuscript and address transparency issues to enable readers to evaluate whether the conclusions of the study are supported by the data has not been adhered to. In the current state, it is extremely difficult to gauge the public health impact of the work

Reviewers' comments:

Reviewer's Responses to Questions

**Comments to the Author**

1. If the authors have adequately addressed your comments raised in a previous round of review and you feel that this manuscript is now acceptable for publication, you may indicate that here to bypass the “Comments to the Author” section, enter your conflict of interest statement in the “Confidential to Editor” section, and submit your "Accept" recommendation.

Reviewer #2: (No Response)

2. Is the manuscript technically sound, and do the data support the conclusions?

Reviewer #2: Partly

3. Has the statistical analysis been performed appropriately and rigorously? 

Reviewer #2: Yes

4. Have the authors made all data underlying the findings in their manuscript fully available?

Reviewer #2: Yes

5. Is the manuscript presented in an intelligible fashion and written in standard English?

Reviewer #2: No

6. Review Comments to the Author

Reviewer #2: The revised manuscript remains difficult to review primarily due to poor organization and formatting of tables. As stated in my previous comments (on the original and R1 version), ideally, the manuscript should have been divided into four separate systematic reviews, with each review discussing the study characteristics, risk of bias, and findings, and including summary tables and figures.

The authors did not address the recommendation by myself and the journal editors to create a summary table for each assessment. Because of inadequate formatting, organization, and presentation of the information, I am not able to review table 3 (findings for all studies combined) or critically review this manuscript. Well-designed summary tables are needed for each assessment to understand the discussion of the results and whether the database of studies supports the authors' conclusions. (See recent IARC monograph tables for examples on presenting results and study characteristics). Reasons include the following:

• The meta-analysis includes only 9 of 24 studies of childhood asthma and formaldehyde; the strength of the evidence of the other 15 studies is not transparent. The supplemental figures mix findings from studies included in the meta-analysis (additional risk estimates) and studies with those not included in the meta-analysis. The scatter plots are not self-explanatory without information from a summary table (e.g., table 3, which is difficult to read). The discussion of the findings of the studies not included in the meta-analysis is inadequate.

• The meta-analysis of childhood asthma exacerbation and symptoms included only 5 of 23 studies. The presentation of results suffers similar problems as discussed above.

• The discussion of the findings from adult studies is more comprehensive than that of the childhood studies not included in the meta-analyses. However, there is no comprehensive resource for the findings across studies in adults (except in Table 3, which is hard to read). Results from the studies are reported in multiple figures that are in the supplement files and thus not reader-friendly. The authors note the variety of exposure categories made it challenging to easily compare across different studies, thus, it is unclear why the authors are reluctant to make changes to improve the presentation of the data.

I appreciate that the authors address several comments made in my previous review. However, I feel the authors ignored (e.g., did not provide a rebuttal or make changes) other comments by either not extracting them into the response document or mistakenly stating they addressed them (e.g., Table 3 was not divided into 4 separate tables).

As the manuscript has the potential to have a public health impact, it is disappointing that the authors are reluctant to improve its readability and transparency to enable readers to evaluate whether the science supports their conclusions.

7. PLOS authors have the option to publish the peer review history of their article (what does this mean?). If published, this will include your full peer review and any attached files.

Reviewer #2: No

- - - - -

---

## [Author Response · Author response to Decision Letter 2]

1 Dec 2020

Our responses to reviewers can be found in the excel file attached and in the cover letter.

---

## [Editor Report · Decision Letter 3]

18 Feb 2021

PONE-D-19-21348R3

Exposure to Formaldehyde and Asthma Outcomes: A Systematic Review, Meta-Analysis, and Economic Assessment

PLOS ONE

Dear Dr. Woodruff,

Thank you for submitting your manuscript to PLOS ONE. After careful consideration, we feel that it has merit but does not fully meet PLOS ONE’s publication criteria as it currently stands. Therefore, we invite you to submit a revised version of the manuscript that addresses the points raised during the review process.

In order to improve the readability of the article, please move tables 3a and 3b to the supplementary material. In addition, please consider dividing table 3b by study design.

We look forward to receiving your revised manuscript.

Kind regards,

A. Kofi Amegah, PhD

Academic Editor

PLOS ONE

Davor Plavec

Academic Editor

PLOS ONE

Additional Editor Comments (if provided):

My concern still relates with Table 3b which is very lengthy (193 pages) and will be extremely difficult for readers to digest this table which presents information on study characteristic of the included studies. The authors should consider breaking down this table according to Study Design (i.e. Cohort, Cross-sectional etc.) to improve the readability

---

## [Author Response · Author response to Decision Letter 3]

22 Feb 2021

All of our comments can be found in our cover letter.

---

## [Editor Report · Decision Letter 4]

24 Feb 2021

Exposure to Formaldehyde and Asthma Outcomes: A Systematic Review, Meta-Analysis, and Economic Assessment

PONE-D-19-21348R4

Dear Dr. Woodruff,

We’re pleased to inform you that your manuscript has been judged scientifically suitable for publication and will be formally accepted for publication once it meets all outstanding technical requirements.

Kind regards,

A. Kofi Amegah, PhD

Academic Editor

PLOS ONE
---

## [Editor Report · Acceptance letter]

5 Mar 2021

PONE-D-19-21348R4 

Exposure to Formaldehyde and Asthma Outcomes: A Systematic Review, Meta-Analysis, and Economic Assessment 

Dear Dr. Woodruff:

I'm pleased to inform you that your manuscript has been deemed suitable for publication in PLOS ONE. Congratulations! Your manuscript is now with our production department. 

Kind regards, 

on behalf of

Dr. A. Kofi Amegah 

Academic Editor

PLOS ONE